# Towards Robust Scale-Invariant Mutual Information Estimators

**Cheuk Ting Leung**[*]                                                      *cheukting.leung@u.nus.edu*
*Department of Electrical and Computer Engineering*
*College of Design and Engineering*
*National University of Singapore*

**Rohan Ghosh**[*]                                                          *rghosh92@gmail.com*
*Department of Electrical and Computer Engineering*
*College of Design and Engineering*
*National University of Singapore*

**Mehul Motani**                                                            *motani@nus.edu.sg*
*Department of Electrical and Computer Engineering*
*College of Design and Engineering*
*Institute of Data Science*
*N.1 Institute for Health*
*Institute for Digital Medicine (WisDM)*
*National University of Singapore*

**Reviewed on OpenReview:** *https://openreview.net/forum?id=vB7Wvytko5*

## Abstract

Mutual information (MI) is hard to estimate for high dimensional data, and various estimators have been proposed over the years to tackle this problem. Here, we note that there exists another challenging problem, namely that many estimators of MI, which we denote as $I(X;T)$, are sensitive to scale, i.e., $I(X;\alpha T) \neq I(X;T)$ where $\alpha \in \mathbb{R}^+$. Although some normalization methods have been hinted at in previous works, there is no in-depth study of the problem. In this work, we study new normalization strategies for MI estimators to be scale-invariant, particularly for the Kraskov–Stögbauer–Grassberger (KSG) and the neural network-based MI (MINE) estimators. We provide theoretical and empirical results and show that the original un-normalized estimators are not scale-invariant and highlight the consequences of an estimator's scale-dependence. We propose new global normalization strategies that are tuned to the corresponding estimator and scale invariant. We compare our global normalization strategies to existing local normalization strategies and provide intuitive and empirical arguments to support the use of global normalization. Extensive experiments across multiple distributions and settings are conducted, and we find that our proposed variants KSG-Global-$L_\infty$ and MINE-Global-Corrected are most accurate within their respective approaches. Finally, we perform an information plane analysis of neural networks and observe clearer trends of fitting and compression using the normalized estimators compared to the original un-normalized estimators. Our work highlights the importance of scale awareness and global normalization in the MI estimation problem.

---

[*]Equal Contribution.

# 1 Introduction

Mutual information (MI), is a fundamental measure of dependency between two variables, which has become pivotal in various machine learning domains, including generalization (Xu & Raginsky, 2017; Bu et al., 2019; Russo & Zou, 2020), representation learning (Bachman et al., 2019; Tschannen et al., 2020) and fairness (Wang et al., 2023; Roh et al., 2020). Estimating MI for high-dimensional continuous variables (Xu et al., 2020) is particularly challenging, due to the hardness of accurately estimating the probability distribution in high dimensions (Goldfeld & Greenewald, 2021). For example, traditional estimators like Kraskov–Stögbauer–Grassberger (KSG) (Kraskov et al., 2004), rely on distance metrics, and for high dimensional data, the distances would have less variation due to the curse of dimensionality.

In this paper, we highlight an important but underexplored factor affecting MI estimation accuracy: the scale of the variables ($|X|$). Specifically, for mutual information $I(X; \alpha T)$, where $\alpha \in \mathbb{R}^+$ is a scaling factor, we show that MI estimates become dependent on $\alpha$ and tend to converge to low values at both extremes ($\alpha \to 0^+$ and $\alpha \to \infty$). This is problematic since, by definition, $I(X; \alpha T) = I(X; T)$ for any two continuous random variables (RVs), and more generally, $I(X; f(T)) = I(X; T)$ for any continuous and invertible transformation $f$ (Cover & Thomas, 2006). Our study focuses specifically on the impact of scale, revealing its significant role in MI estimation errors.

As we show in this work, commonly used MI estimators such as binning, KSG (Kraskov et al., 2004) and MINE (Belghazi et al., 2018) are not natively scale invariant. Typically this is addressed via normalization approaches. Despite numerous surveys that have explored various methods of MI estimation, some examples of which include (Walters-Williams & Li, 2009; McAllester & Stratos, 2020; Paninski, 2003), an in depth study of normalization approaches has been absent. A standard approach used in most works is *local normalization*, where each dimension is adjusted to have a variance of 1 (Hjelm et al., 2019; Xie et al., 2024; Kraskov et al., 2004). However, local normalization treats each dimension independently and normalizes them to have a variance of 1, which, as we demonstrate in Section 4.1.1, does not work well in the high-dimension setting especially in neural networks, across two separate experiments. This is because most high-dimensional feature representations in neural networks always contain some noisy dimensions, which are of low energy and contain irrelevant features. We rigorously verify this in Appendix J. Thus, amplifying these low energy dimensions can lead to suboptimal MI estimates. We also note that the recent work by (Czyż et al., 2023), in addition to trying out local normalization approaches, also studied other preprocessing methods including the transformation of the margin distribution to uniform distribution (via converting to rank). We note that this conversion step also brings all individual dimensions to equal importance like local normalization, and thus would have the same pitfalls in this scenario.

To address this issue, in our work, we propose a set of *global normalization* approaches. Unlike local normalization, global normalization preserves the relative energies between the different dimensions, and thus avoids scaling up low-energy noisy dimensions. Our proposed estimator modifications do not only include new normalization approaches, however, and often also have an additional maximization step, which helps bias our estimators better. It is well known that KSG and other MI estimators have a tendency to have negative bias Czyż et al. (2023), especially in high dimensions. Our normalization approaches for KSG incorporate this observation via an additional maximization step, which also follows intuitively from one of our theoretical observations in Proposition 3.

We summarize our contributions as follows:
- We theoretically show that native KSG and MINE estimators depend on the scale of random variables.
- We propose scale-invariant KSG and MINE extensions that resolve one-sided scale issues and significantly improve accuracy. Our analysis of normalization methods is, to our knowledge, the first of its kind.
- We show that KSG-Global-$L_\infty$ and MINE-Global-Corrected consistently yield the most accurate MI estimates across varied synthetic experiments, including high-dimensional, low-data setups.
- We analyze MI dynamics between inputs and neural network layer outputs during training, finding that unnormalized estimators confound scale, whereas our methods reveal distinct training phases such as fitting and compression.

Note that a detailed background on MI Estimators studied in this work is presented in Appendix A. Next, we provide a summary of related works that have either studied robustness of MI estimators to transformations,

or proposed transformation robust MI estimators. Note that a more detailed discussion of what follows is available in Appendix B.

## 1.1 Overview of Related Works

The notion of self-consistent equitability, discussed in (Reshef et al., 2013; Kinney & Atwal, 2014), is a more general study of invariance of MI estimators, but has not explicitly studied scaling due to preprocessing techniques enforcing scale invariance. These works assess invariance in MI estimates across a broad range of transformations, but our study differs by prioritizing MI estimation accuracy and testing cascaded transformations that modify dimensionality and structure. Unlike previous studies that apply one-dimensional transformations independently to each feature, our approach introduces richer transformations that could be further explored within an equitability framework. Similarly, recent MI estimation studies such as (Czyż et al., 2023) mainly focus on dimension-wise transformations, whereas we consider transformations relevant to deep learning applications, such as sigmoid activation and random matrix multiplications. Preprocessing wise, most works have focused on standard local normalization. While (Czyż et al., 2023) finds Gaussianization-based local normalization beneficial for long-tailed distributions, the improvements are minor and they end up preferring standard local normalization. Notably, scale invariance is absent in their analysis due to preprocessing techniques that normalize local scales, whereas our study explicitly ensures scale-invariant MI estimates while maintaining robustness to noisy dimensions.

Neural network-based preprocessing strategies for MI estimation have also been explored in recent works. Gowri et al. (2024) propose a two-step MI estimation method where compressed representations are learned to minimize an upper bound on conditional entropy before applying KSG estimation. While their approach inherently involves scale variations due to neural network compression, they mitigate this using local unit normalization, and our findings on global normalization strategies can be applied to their framework to potentially enhance performance. Similarly, Butakov et al. (2024) introduce an MI estimator based on normalizing flows, transforming data distributions into simpler forms with closed-form MI solutions. Their method primarily focuses on Gaussian base distributions, and while it may be inherently scale-invariant, no explicit theoretical analysis has confirmed this. Our work highlights the importance of scale invariance in MI estimation, emphasizing on preventing scale confounding in estimator outputs.

## 2 Motivation

Estimating MI is fundamental to various domains, ranging from learning theory to practical applications such as medical analysis and wireless communication (Shwartz-Ziv & Tishby, 2017; Saxe et al., 2018). To motivate our proposed normalization strategy, this section outlines several desirable properties that effective MI estimators should possess. Let $S = \{(X_1, T_1), (X_2, T_2), \ldots, (X_n, T_n)\}$ be the sampled data. With this, let $\widehat{I}_{est}^n(X; T)$ represent an estimate of the MI between $X$ and $T$ using the estimator $est$, given $n$ sampled points from the joint distribution $P(X, T)$. Ideally, we seek the estimator to have the following properties:

1. **Global Scale Invariance:** For any $\alpha \in \mathbb{R}^+$ and $n \in \mathbb{Z}^+$, $\widehat{I}_{est}^n(\alpha X; \alpha T) = \widehat{I}_{est}^n(X; T)$
2. **One-Sided Scale Invariance** For any $\alpha \in \mathbb{R}^+$ and $n \in \mathbb{Z}^+$, $\widehat{I}_{est}^n(X; \alpha T) = \widehat{I}_{est}^n(X; T)$

We emphasize the importance of these properties because true mutual information inherently satisfies them. By definition, $I(\alpha X; \alpha T) = I(X; T)$ and $I(\alpha X; T) = I(X; T)$ for a scalar $\alpha$. In the case of neural networks, where $X$ represents the input, $Y$ represents the target, and $T$ represents the features, estimation of $I(X; T)$ becomes important, as it was hypothesized that it can predict the generalization behavior of deep learning networks (Shwartz-Ziv & Tishby, 2017). Furthermore, (Shwartz-Ziv & Tishby, 2017) also predicts a two-phase behavior of $I(X; T)$ during training: (a) fitting, where $I(X; T)$ and $I(T; Y)$ increases, and (b) compression where $I(X; T)$ decreases. However, this is often not observed (Saxe et al., 2018). We hypothesize that it could be because of the scale-sensitivity of the estimators, as the scale of $T$ changes significantly during training.

We note that the current estimators may not obey one-sided scale invariance. First, we study three estimators theoretically: KSG, MINE, and binning.

## 2.1 Testing One-sided Scale-Invariance of MI Estimators

In this section, we introduce new theoretical results that analyze the global-scale invariant and one-sided scale invariant properties of commonly used MI estimators. Note that for all results that follow, we assume every RV is bounded. That is, if $X$ is bounded, we have that $|X| \leq B$ for some finite $B < \infty$. Also, for the following results, let $X \in \mathbb{R}^d$ and $T \in \mathbb{R}^m$. Also note that by $\mathbb{R}^+$, we mean the set of all positive real numbers, excluding zero.

**Binning:** Let us denote the binning estimator described in (Paninski, 2003) by $\widehat{I}^n_{bin}$. Then we have the following result.

**Proposition 1.** Let $\widehat{I}^n_{\text{bin}}(X;T)$ denote the mutual information estimated using a fixed number of bins per dimension, with bin edges defined by the minimum and maximum values of the data. Then, for all $\alpha \in \mathbb{R}^+$,

$$\widehat{I}^n_{\text{bin}}(\alpha X; \alpha T) = \widehat{I}^n_{\text{bin}}(X;T) \quad \& \quad \widehat{I}^n_{\text{bin}}(X; \alpha T) = \widehat{I}^n_{\text{bin}}(X;T).$$

**Remark 1.** We note that even though the binning estimator is scale-invariant, the native binning estimator is not a good estimator for MI, more so in the high dimension setting (Kraskov et al., 2004). This is because in high dimensions the data occupies the space sparsely, and most bins will yield zero datapoints and thus a zero probability. However, in recent years there have been variants proposed based on structured density estimation, which are more robust for use in high dimensional spaces. Nonetheless, we only investigate the KSG and MINE estimators in our work. Additional discussions on state-of-the-art binning estimators is provided in Appendix A.

**KSG:** Let us denote the KSG estimator proposed in (Kraskov et al., 2004) by $\widehat{I}^n_{KSG}$. Then, we have the following results.

**Proposition 2.** It holds that $\widehat{I}^n_{KSG}(\alpha X; \alpha T) = \widehat{I}^n_{KSG}(X;T), \forall \alpha \in \mathbb{R}^+$.

**Proposition 3.** Let $\{(X_i, T_i)\}_{i=1}^n$ be drawn i.i.d. from a bounded distribution on $\mathbb{R}^{d_X} \times \mathbb{R}^{d_T}$ that is absolutely continuous. Then, it holds almost surely that $\lim_{\alpha \to 0^+} \widehat{I}^n_{KSG}(X; \alpha T) = \lim_{\alpha \to \infty} \widehat{I}^n_{KSG}(X; \alpha T) = -\frac{1}{k}$. Thus, $\widehat{I}^n_{KSG}(X; \alpha T)$ need not be equal to $\widehat{I}^n_{KSG}(X;T)$.

**MINE:** We first define two variants of the MINE estimator as follows:

**MINE-Opt:** This estimator refers to the MINE estimator where instead of training the neural network on MINE's loss function defined in 7 by stochastic gradient descent (SGD), we pick the best neural network configuration that directly maximizes 7. Thus, we pick the global optimum.

**MINE-SGD:** This estimator refers to the MINE estimator where optimization of the loss function defined in 7, is performed using conventional stochastic gradient descent. This is the standard approach proposed originally by (Belghazi et al., 2018).

We denote the MINE-based MI estimators by $\widehat{I}^n_{MINE-opt}$ and $\widehat{I}^n_{MINE-sgd}$. We then have the following results.

**Proposition 4.** It holds that $\widehat{I}^n_{MINE-opt}(X; \alpha T) = \widehat{I}^n_{MINE-opt}(X;T) \ \forall \alpha \in \mathbb{R}^+$.

Next, we outline a theoretical result regarding the limiting behaviour of the first layer weights for the MINE estimator's neural network, when the scale of one of the variables approaches zero.

**Proposition 5.** Consider the MINE optimization problem with input data $S = \{(\alpha X_1, Y_1), \dots, (\alpha X_n, Y_n)\}$ where $X \in \mathbb{R}^{d_x}$, $Y \in \mathbb{R}^{d_y}$, $(X, Y) \sim P(X, Y)$ are bounded RVs and $\alpha \in \mathbb{R}^+$ is a scaling factor. We consider a neural network of depth $L + 1$ having $h_1, h_2, \dots, h_L$ ReLU-activated hidden neurons in the respective layers. The network is trained via gradient descent on the MINE loss function in Belghazi et al. (2018) for a fixed number of iterations $n_T$, with a learning rate schedule $0 \leq \eta(t) < \infty$ for all $t \leq n_T$. Let the weights between the $i^{th}$ node of the $l + 1^{th}$ hidden layer and the $j^{th}$ node of the $l^{th}$ hidden layer after $t$ iterations be denoted by $w^l_{ji}(t)$. Assume that the initialized weights are bounded, i.e., $\forall (l, i, j), |w^l_{ji}(0)| \leq \epsilon$ for some $\epsilon > 0$. Lastly, let $w^0_{ji}[X]$ denote the first layer weights that are attached to $X$. We then have, $\forall (i, j)$,

$$\lim_{\alpha \to 0^+} \left| w^0_{ji}[X](n_T) \right| \leq \epsilon. \tag{1}$$

With this, we have the following result that explores scale invariance for MINE.

**Proposition 6.** We consider the same setting as Proposition 5 for the MINE estimation problem. There, it holds that $\lim_{\alpha \to 0^+} \widehat{I}^n_{MINE-sgd}(X; \alpha T) = 0$ . Thus, $\widehat{I}^n_{MINE-sgd}(X; \alpha T)$ need not be equal to $\widehat{I}^n_{MINE-sgd}(X; T)$.

## 3 Methodology

### 3.1 Normalization Strategies

We consider a setting where we are given an RV $X = [x_1, x_2, \ldots, x_d] \in \mathbb{R}^d$, where $x_i \in \mathbb{R}$ represents the $i$-th component of $X$. Suppose $X \sim P$, where $P$ is a probability distribution, and let $S = \{X_1, X_2, \ldots, X_n\}$ be an independent and identically distributed (i.i.d.) sample drawn from $P$, with $X_j \in \mathbb{R}^d$ for $j = 1, \ldots, n$. With this, we outline three normalization strategies that form the basis of our studies in this work. We define them as follows.

**Definition 1. (Local Normalization)** The *locally normalized variable* $X_{\sigma|S} = [x'_1, \ldots, x'_i, \ldots, x'_d] \in \mathbb{R}^d$ is defined by normalizing each dimension $i$ individually as $x'_i = \dfrac{x_i - \bar{x}_i}{\sqrt{\frac{1}{n} \sum_{j=1}^n (x_{i,j} - \bar{x}_i)^2}}, \quad$ for $i = 1, \ldots, d$, where

the empirical mean is defined as: $\bar{x}_i = \frac{1}{n} \sum_{j=1}^n x_{i,j}$.

**Definition 2. (Global Normalization)**
The *globally normalized variable* $X_{\Sigma|S} \in \mathbb{R}^d$ is defined as $X_{\Sigma|S} = \dfrac{X - \bar{X}}{\sqrt{\frac{1}{n} \sum_{j=1}^n \|X_j - \bar{X}\|_2^2}}$, where $\|\cdot\|_2$ denotes the

$L_2$-norm, and $\bar{X}$ is the empirical mean: $\bar{X} = \frac{1}{n} \sum_{j=1}^n X_j$.

**Definition 3. (Global $L_\infty$ Normalization)**
The *globally $L_\infty$-normalized variable* $X_{\Sigma_\infty|S} \in \mathbb{R}^d$ is $X_{\Sigma_\infty|S} = \dfrac{X - \bar{X}}{\frac{1}{n} \sum_{j=1}^n \|X_j - \bar{X}\|_\infty}$, where $\|\cdot\|_\infty$ denotes the

$L_\infty$-norm and $\bar{X}$ is the empirical mean: $\bar{X} = \frac{1}{n} \sum_{j=1}^n X_j$.

Note that for any RV $X$, we denote by $X_{\sigma|S}$ and $X_{\Sigma|S}$ its locally and globally normalized versions respectively.

### 3.2 Studied Scale-Invariant Estimators

We are given the RVs $X \in \mathbb{R}^d$ and $T \in \mathbb{R}^m$, and sampled data $S = \{(X_1, T_1), (X_2, T_2), \ldots, (X_n, T_n)\} \sim P^n_{XT}$. All following estimates are for the MI between $X$ and $T$, given $S$. With this, we propose the following normalization approaches for KSG and MINE estimators. We outline our approaches for scale-invariant KSG and MINE extensions in Table 1.

Table 1: Proposed Scale-Invariant KSG and MINE variants

| KSG | MINE |
|---|---|
| **KSG-Local**: $\hat{I}_{KSG}(X_{\sigma|S}; T_{\sigma|S})$ | **MINE-Local**: $\hat{I}_{MINE}(X_{\sigma|S}; T_{\sigma|S})$ |
| **KSG-Global**: $\max\limits_{c \in \{c_1, c_2, \ldots, c_m\}} \left[ \hat{I}_{KSG}(X_{\Sigma|S}; cT_{\Sigma|S}) \right]$ | **MINE-Global**: $\hat{I}_{MINE}(X_{\Sigma|S}; T_{\Sigma|S})$ |
| **KSG-Global-$L_\infty$**: $\max\limits_{c \in \{c_1, c_2, \ldots, c_m\}} \left[ \hat{I}_{KSG}(X_{\Sigma_\infty|S}; cT_{\Sigma_\infty|S}) \right]$ | **MINE-Global-Corrected**: $\hat{I}_{MINE}(\sqrt{d_X} X_{\Sigma|S}; \sqrt{d_T} T_{\Sigma|S})$ |

**Remark 2.** In addition to the above approaches, we compare the standard baselines of KSG and MINE. Furthermore, we also include a updated variant of KSG in our comparisons, called BI-KSG (Gao et al., 2017), which has smaller bias levels for highly correlated data. We do not include the native binning estimator in our experimental results, as we find that they fare poorly for almost all of our studied cases. Thus, we only study the KSG and MINE variants empirically in this work. Also, note that the range of multiplier scales $c_1, c_2, \ldots, c_m$ are tunable hyperparameters, and we fix $c_1 = 0.1, c_2 = 0.2, \ldots, c_m = 2$ for all our experiments. Note that for the KSG-Global variants, this does slow down the MI estimation process, as it takes $m$ times the computation time to estimate the measure when compared to KSG.

## 4 Experimental Studies

Our empirical studies can be categorized into roughly four broad sections:

1. **Empirical motivation for proposed normalization variants (E1-E3):** We provide in-depth empirical analyses for each normalization variant proposed in this work, and also the overall reasons for potentially choosing global normalization approaches over local ones.
2. **Scale dependence and Signal to Noise Ratio (SNR) analysis of estimators (E4,E5):** We perform some basic tests and analyses of all estimators. First, we study their overall responses to scale changes, and then we study their responses to changes in noise levels.
3. **Accuracy analysis of estimators (E6):** We conduct an extensive accuracy-bias-correlation analysis of all estimators in three different settings where ground truth MI is known. In each setting, we generate synthetic data using a diverse set of transformations to simulate different distribution scenarios.
4. **Studying neural network training using estimators (E7):** We study the MI dynamics of neural networks during training. Specifically, we analyze the MI between inputs and features and compare the trends resulting from various estimators.

Note that the experiments are numbered from **E1** to **E7** for clarity. We use three *base* distributions for generating the RVs $X$ and $T$. We refer to them in various parts of the experiments. They are:

- **Correlated Gaussians:** Here, we consider a joint Gaussian distribution where $(X, T) \in \mathbb{R}^d \sim \mathcal{N}(0, \Sigma)$ with covariance matrix:
$$\Sigma = \begin{bmatrix} I_d & \rho I_d \\ \rho I_d & I_d \end{bmatrix}.$$
This ensures that $\mathbb{E}[X_i T_i] = \rho$ for $1 \le i \le d$ while $\mathbb{E}[X_i T_j] = 0$ for $i \ne j$.
- **Additive Gaussian Noise:** Here $X \in \mathbb{R}^d \sim \mathcal{N}(0, I_d)$ and $T = X + \epsilon$, where $\epsilon \sim \mathcal{N}(0, \sigma^2 I_d)$.
- **Correlated Student's t-distributions:** We sample correlated $X, T \in \mathbb{R}^d$ from a multivariate Student's t-distribution. Details on the sampling procedure are provided in Appendix D (same as (Czyż et al., 2023)).

The details for our estimators are provided in **Appendix** F. We used the NPEET MI estimator toolbox for estimating KSG and KSG-based measures [1]. For MINE, we used a pytorch-based package [2]. Code for all our experiments is available in the Supplementary Material.

### 4.1 E1-E3: Additional Motivation for Normalization Variants

In this section, we provide both intuitive and empirical arguments for every aspect of our proposed variants in the previous section: (**E1**) why global normalization and not local, (**E2**) why maximization of MI for KSG and (**E3**) why the specific choice of global normalization variants: KSG-Global-$L_\infty$ and MINE-Global-Corrected.

#### 4.1.1 E1: Global Over Local

Our main observation is that local normalization scales all variables equally, potentially overemphasizing irrelevant, low-energy dimensions, which degrades MI estimates. However, global normalization preserves the distance structure of the original RVs, which can benefit in high-dimensional settings, such as neural network feature representations, where many features in $T$ are sparse. We highlight specific cases for KSG and MINE below, where local normalization yields undesirable behaviour, which is avoided by global normalization.

**KSG:** Consider RVs $X, T \in \mathbb{R}^2$, where $T = X + \epsilon$ with $\epsilon \sim \mathcal{N}(0, \sigma^2 I_2)$. We augment $X$ with $k$ independent noise components $\epsilon = [\epsilon_1, \dots, \epsilon_k]$, $\epsilon_i \sim \mathcal{N}(0, \sigma'^2)$ with $\sigma' \ll \sigma$, forming $X' = [X, \epsilon]$. Although $I(X; T) = I(X'; T)$, as shown in Figure 1a, KSG and global normalization maintain stable MI estimates with increasing $k$, while local normalization underestimates MI by over-scaling the noise.

**MINE:** For MINE, we generate correlated Gaussian RVs $X, T \in \mathbb{R}^2$ (with a random correlation $\rho \in (0, 0.8)$) and similarly extend $X$ with noise to form $X'$. Averaging over 10 trials, Figure 1b reveals that local

---

[1] https://github.com/gregversteeg/NPEET    [2] https://github.com/gtegner/mine-pytorch

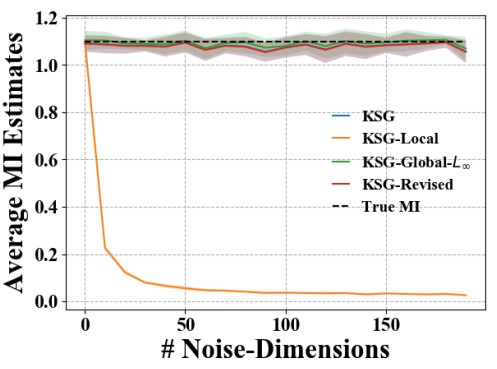 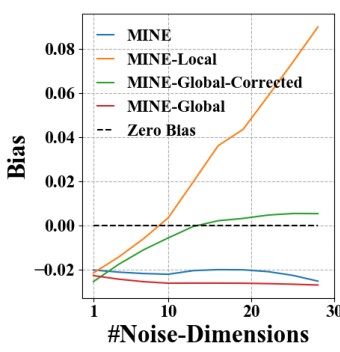

(a) Average MI estimates ($I(X;T')$) for KSG-based measures for varying noise dimensions ($k$).

(b) Bias of MINE-based measures for varying noise dimensions.

Figure 1: (**E1**) Comparative evaluation of MI estimators (KSG and MINE) under increasing noise dimensions. MI should be unchanged, but local normalization significantly impacts estimates in both cases.

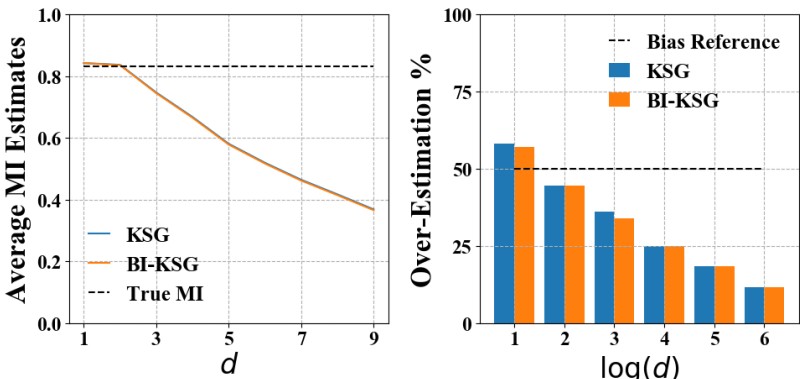

Figure 2: (**E2**) Dependency of KSG estimator bias on data dimension (using base-2 log). We find that as dimensionality increases, KSG increasingly underestimates the true MI.

normalization causes the bias to increase significantly with $k$, whereas global normalization yields stable estimates. This occurs because, for MINE, the network tends to overfit the added noise when each variable is normalized equally, while global variants preserve the effective input dimensionality.

### 4.1.2 E2: KSG-Global: Why the Maximization Step?

We use a maximization step in our global normalization estimator for KSG in Table 1. We summarize two main arguments for our maximization step:

1. **Negative Bias in High Dimensions:** The KSG estimator shows increasing negative bias as data dimensionality grows. In one experiment, correlated Gaussians $X, T \in \mathbb{R}^d$ with fixed ground truth MI ($\approx 0.8$) were sampled (20 trials of 1000 points each) while $d$ ranged from 1 to 9, yielding a clear negative bias in the estimators as seen in Figure 2(a). A second experiment with $d \in \{2, 4, 8, 16, 32, 64\}$ and randomly chosen correlation $\rho$ confirmed that the fraction of MI estimates below the ground truth rises significantly with $d$ (Figure 2(b)). Taking the maximum estimate over scales helps mitigate this bias.

2. **Consequence of Proposition 3:** Proposition 3 shows that for $I(X; \alpha T)$, the KSG estimator converges to negative values at extreme scales $\alpha$. This observation motivates taking the maximum over a range of scales $c$ for the globally normalized variables, i.e., $\widehat{I}^n_{KSG}(X_{\Sigma|S}, cT_{\Sigma|S})$. Empirically, the estimator values exhibit an approximately Gaussian trend with respect to $c$ (see Figure 4a), making the maximum both well-defined and meaningful.

**Remark 3.** Note that MINE implicitly maximizes over relative scales via the arbitrary first-layer weights. Since scaling these weights adjusts the effective input similarly (i.e., $(\alpha W)^T X = W^T(\alpha X)$), the network optimizes over affine transformations. Nonetheless, due to the tendency of gradient descent towards flatter minima Keskar et al. (2017), the full benefit of this invariance may not be achieved.

### 4.1.3 E3: Motivation for Global Normalization Variants

Next, we motivate our proposed global normalization variants: KSG-Global-$L_\infty$ and MINE-Global-Corrected.

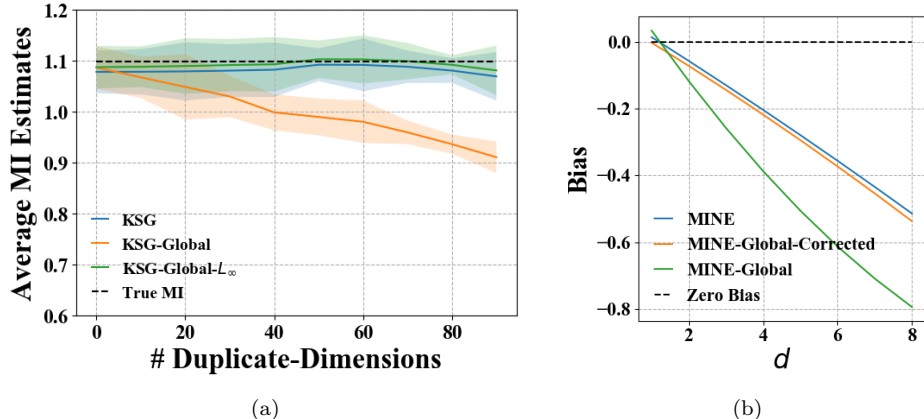

(a)                                        (b)

Figure 3: (**E3**) Analysis of normalization variants: (a) Impact of data duplication for KSG-based approaches, and (b) Estimator bias for MINE and MINE-Global variants across dimensions. These results highlight the importance of our proposed global normalization variants.

**KSG:** We observe that since KSG computes distances using the $L_\infty$-norm, when $d_X \gg d_T$, global normalization makes the individual dimensions of $X$ significantly smaller than those of $T$, yielding smaller $L_\infty$ distances. This motivates the proposed KSG-Global-$L_\infty$ approach, which scales based on $L_\infty$ norm distances instead of $L_2$ norm. To illustrate its impact, we duplicate $X$ to form $X' = [X, X, \ldots, X]$ (thus $I(X'; T) = I(X; T)$). As shown in Figure 3a, KSG-Global actually decreases its estimate with increasing duplicates, whereas both KSG and KSG-Global-$L_\infty$ maintain consistent MI estimates. This confirms the necessity of employing the $L_\infty$-norm for scaling the variables using the global normalization approach.

**MINE:** Global normalization can lead to low per-dimension energy when $d_X \gg d_T$, since $\mathbb{E}[X_{\Sigma|S}(i)^2] = 1/d_X$ versus $\mathbb{E}[T_{\Sigma|S}(i)^2] = 1/d_T$. This imbalance may cause gradient descent to focus predominantly on $T$. To counter this, we rescale the normalized variables so that $\mathbb{E}[X'_{\Sigma|S}(i)^2] = \mathbb{E}[T'_{\Sigma|S}(j)^2] = 1$, i.e., $X'_{\Sigma|S} = \sqrt{d_X}\, X_{\Sigma|S}$ (and similarly for $T$). Experiments with $d$ ranging from 1 to 9 (Figure 3b) reveal that while MINE-Global's bias grows more negative with higher $d$, MINE-Global-Corrected has bias levels comparable to standard MINE. This demonstrates that rescaling effectively mitigates the effects of imbalances in terms of per-dimension energy of $X$ and $T$.

## 4.2 (E4-E5) Scale and SNR Analysis

In this analysis, we test the scale invariance properties of all estimators, and subsequently test their response to changes in noise levels (SNR).

**Scale:** Experiments were conducted to study the effect of scaling on MI estimators $I(\eta X; T)$, using correlated Gaussian variables $X, T$. Scaling factors $\eta$ were sampled logarithmically between $10^{-2}$ and $10^3$ for KSG, and $10^{-2}$ to 10 for MINE, as MINE estimates degrade sharply beyond $\eta = 10$. Results (Figure 4a, 4b) show KSG estimates converging to $-0.33$ (matching Proposition 3) as $\eta$ increases, while MINE estimates drop to zero

as $\eta \to 0$ (supporting Proposition 6). Global and local normalization variants remained robust to scaling, with significantly lower RMSE compared to vanilla estimators (Figure 4c, 4d).

**SNR:** MI estimators were tested with Signal-to-Noise Ratios (SNR) ranging from 0 to 5, where $T = X + \epsilon$ ($\epsilon \sim \mathcal{N}(0, \sigma^2)$) and scaled to $T' = 0.1T$ (estimating $I(X; T')$). We expect the MI estimates to increase with SNR, following the ground truth. Scale invariance could be important here, as the noise variance $\sigma$ and the final scale of $T'$ will be related, and thus any estimator bias accompanying large scales can adversely affect the trend. Results (Figure 5a, 5b) show global and local variants accurately following ground truth MI trends, increasing with SNR. However, vanilla KSG and MINE estimators failed to reflect the true MI trend, with MINE estimators even stabilizing at higher SNR values due to their inherent scale dependence.

### 4.3 E6: Comparing MI Estimators: Error Analysis

In this section, we undergo a comprehensive series of experiments, where we compute various error measures of all estimators on a diverse range of datasets.

#### 4.3.1 Experiment Summary

**Dataset creation:** To create these datasets, we follow the three base distributions described in the beginning of this section. First, we generate $X, T$ according to the base distributions: Additive Gaussian, Correlated Gaussian and Correlated Student's-t as defined in Section 4. Then, we then make $X$ undergo some (or none) of the following transformations, which are all MI preserving. For what follows, let $X \in \mathbb{R}^d$ and $T \in \mathbb{R}^d$.

1. **Randmat (rm):** $X' = \alpha W^T X$, where $\alpha \sim Unif(0,1)$ and $W \in \mathbb{R}^{d \times d}$ where $W(i,j) \sim Unif(0,1)$. $Unif(a,b)$ denotes a uniform distribution over $[a,b]$. If the randomly generated $W$ is not invertible, we repeat the sampling process until we get an invertible $W$.
2. **Cube (cb):** $X' = X \circ X \circ X$, where $\circ$ denotes element wise multiplication (Hadamard Product).
3. **Sigmoid (sg):** $X' = \sigma(X)$, where $\sigma : \mathbb{R}^d \to \mathbb{R}^d$ is such that $X'[i] = \frac{1}{1+e^{-X[i]}}$, where $X[i]$ denotes the $i^{th}$ dimension of $X$ and similarly for $X'$.
4. **Duplicate-self (ds):** $X' = [X, X, \dots, X] \in \mathbb{R}^{Kd}$. We set $K = 20$ in our experiments.

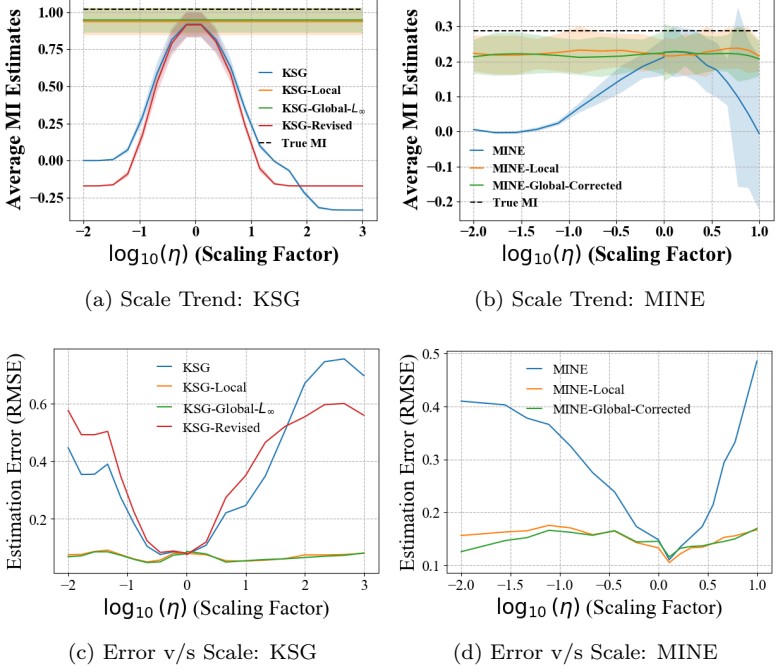

(a) Scale Trend: KSG  (b) Scale Trend: MINE

(c) Error v/s Scale: KSG  (d) Error v/s Scale: MINE

Figure 4: (**E4**) Analysis of MI Estimators in response to data scaling. Estimates are for $I(\eta X; T)$, where $\eta$ is the scaling factor. These results clearly demonstrate the impact of scale on MI estimates.

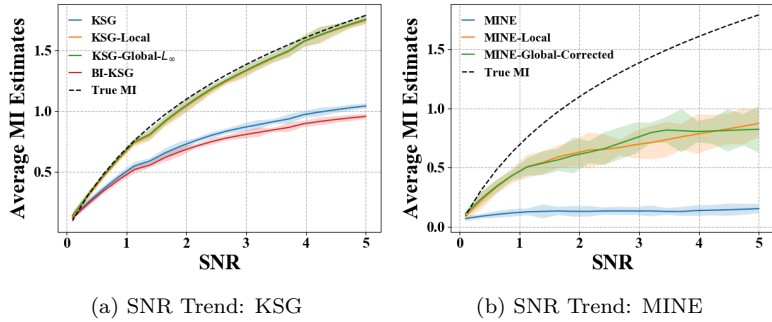

(a) SNR Trend: KSG      (b) SNR Trend: MINE

Figure 5: (**E5**) MI estimates across varying Signal-to-Noise Ratios (SNR). Estimates are for $I(X; \eta T)$, where $T'$ is scaled by $\eta = 0.1$. Normalized variants reflect the SNR trends more accurately, whereas unnormalized estimators may not.

5. **Duplicate-noise (dn):** $X' = [X, \epsilon] \in \mathbb{R}^{d+k}$, where $\epsilon = [\epsilon_1, \epsilon_2, \ldots, \epsilon_k]$ where $\epsilon_i \sim \mathcal{N}\left(0, \sigma'^2\right)$. We set $\sigma' = 0.2$ and $k = 20$.

Our objective is to evaluate the accuracy of the estimation of $I(X'; T)$.

**Performance Measures:** We study three different measures of performance in our experiments. For what follows, let $\hat{\mu}_1, \hat{\mu}_2, \ldots, \hat{\mu}_k$ denote the estimated values of MI for any estimator across $k$ trials, and let $\mu_1, \mu_2, \ldots, \mu_k$ denote the ground truth values. The performance measures used in our evaluation are: (a) **Normalized RMSE:** We first estimate the RMSE as $RMSE(\hat{\boldsymbol{\mu}}, \boldsymbol{\mu}) = \sqrt{\mathbb{E}_i[(\hat{\mu}_i - \mu_i)^2]}$. Then we estimate a baseline RMSE as $RMSE\_Base(\boldsymbol{\mu}) = \sqrt{\mathbb{E}_{i,j}[(\mu_i - \mu_j)^2]}$. With this, we can estimate the final measure as: $RMSE\_Norm(\hat{\boldsymbol{\mu}}, \boldsymbol{\mu}) = \frac{RMSE(\hat{\boldsymbol{\mu}}, \boldsymbol{\mu})}{RMSE\_Base(\boldsymbol{\mu})}$. (b) **Spearman Correlation:** The Spearman correlation measures the degree of monotonic relationship between $\hat{\boldsymbol{\mu}}$ and $\boldsymbol{\mu}$ (Zar, 2005). This is estimated as the Pearson's correlation coefficient between the rank values of $\hat{\boldsymbol{\mu}}$ and $\boldsymbol{\mu}$. (c) **Bias:** We estimate the bias as $\mathbb{E}_i[\mu_i - \hat{\mu}_i]$.

**Evaluation Process:** For each experiment, a specific MI-preserving transformation (listed in the first column of Tables 2 and 3) is applied to $X$. Across 40 trials, we generate $N = 1000$ samples of $X, T \sim P(X, T)$, apply the transformation to obtain $X'$, and estimate $I(X'; T)$. Performance metrics such as normalized RMSE, Spearman correlation, and bias are computed. Red entries indicate cases with RMSE > 1, though they often still exhibit strong Spearman correlation with the true MI. Note that here we only discuss the results for the additive Gaussian noise base (see Appendix G for full results).

Table 2: (**E6**) Normalized RMSE of KSG-Based Estimators: Additive Gaussian Noise Base

| Transformation | | | | | $d$ | KSG-Based Measures | | | | |
|---|---|---|---|---|---|---|---|---|---|---|
| rm | cb | sg | ds | dn | | ksg | bi-ksg | ksg-loc | ksg-glo | ksg-glo-$L_\infty$ |
| | ✓ | ✓ | ✓ | | 2 | 0.351 | 0.404 | 0.147 | 0.208 | **0.110** |
| | | ✓ | ✓ | | 2 | 0.286 | 0.335 | 0.060 | 0.084 | **0.050** |
| | ✓ | ✓ | | | 2 | 0.458 | 0.533 | 0.145 | **0.113** | 0.113 |
| | | ✓ | | | 4 | 1.275 | 1.424 | 0.312 | **0.300** | 0.301 |
| ✓ | | | | ✓ | 4 | 0.862 | 0.932 | 0.445 | 0.594 | **0.396** |
| | | ✓ | | | 4 | 0.332 | 0.342 | **0.304** | 0.520 | 0.297 |
| | | | | ✓ | 4 | 0.332 | 0.342 | 1.327 | 0.298 | 0.297 |
| ✓ | ✓ | | | | 4 | 1.131 | 1.233 | 0.977 | 0.959 | 0.931 |
| | | ✓ | | ✓ | 4 | 1.275 | 1.424 | 1.334 | **0.300** | 0.301 |
| | ✓ | ✓ | | ✓ | 6 | 1.981 | 2.129 | 1.904 | 1.021 | 1.021 |
| | | ✓ | | | 6 | 1.983 | 2.131 | **0.816** | 0.811 | 0.812 |
| | ✓ | | ✓ | | 6 | 1.377 | 1.408 | 1.343 | 1.605 | 1.290 |
| ✓ | | | ✓ | | 6 | 1.643 | 1.730 | 1.275 | 1.660 | 1.153 |
| | | ✓ | | ✓ | 6 | 1.983 | 2.131 | 1.905 | **0.814** | 0.816 |

**Remark 4.** Since every transformation is MI preserving, combining them yields new distributions whose ground truth MI is unchanged. This flexible framework allows us to simulate high-dimensional data (up to 200 dimensions) with low intrinsic dimension (typically <10), reflecting the characteristics of neural network features. Our choice of transformations is motivated by the behaviors observed under different normalization strategies—for example, local normalization is adversely affected by added noise (duplicate-noise), while KSG-Global struggles with duplicate-self, and nonlinear transformations (e.g., sigmoid and cube) can alter nearest neighbor distances.

### 4.3.2 Takeaways

- Overall, global and local normalization variants fare significantly better than the baseline measures.
- Our global normalization variants (MINE-global-corrected and KSG-Global-$L_\infty$) overall fare better than other normalization strategies. In fact, when for the additive Gaussian noise base, we find that in most cases MINE-global-corrected and KSG-global-$L_\infty$ outperform other normalization approaches.
- KSG-Global-$L_\infty$ shows consistent improvements throughout. Even when the normalized RMSE estimates are insignificant (red entries), KSG-Global shows significant correlation with true MI in many of the cases (see Appendix G).
- Aligning with our earlier discussions, we find that overall the global normalization variants (MINE-Global-Corrected and KSG-Global-$L_\infty$) perform better than their vanilla global normalization counterparts. This is much more apparent in the case of MINE.

### 4.4 E7: Application of MI Estimations in Deep Learning

Mutual information is a key measure for analyzing neural network behavior during training. We evaluate MI on IB (Shwartz-Ziv & Tishby, 2017), MNIST (Deng, 2012), CIFAR-10 (Krizhevsky & Hinton, 2009) and SVHN (Netzer et al., 2011) datasets. Network architectures, activations, and other details are in Appendix F. For each classification dataset $\{X, Y\}$, networks are trained and an intermediate layer's output $Z$ is extracted (third layer for IB and MNIST; Global Average Pooling for CIFAR-10 and SVHN). MI estimates $I(X; Z)$ and $I(Z; Y)$ are computed using KSG, KSG-Local, MINE, MINE-Local, and our proposed KSG-Global-$L_\infty$ and MINE-Global-Corrected estimators.

We analyze the MI estimates from two perspectives:

- **Training Dynamics:** In Figure 6, we plot $I(X; Z)$ over epochs (averaged over 10 trials), along with the scale $|Z|$ (in blue). On MNIST and CIFAR-10, the original KSG estimates strongly follow the feature scale, indicating sensitivity to scaling. In contrast, KSG-Global-$L_\infty$ does not mimic the scale curve; for

Table 3: (**E6**) Normalized RMSE of MINE-Based Estimators: Additive Gaussian Noise Base

| Transformation | | | | | $d$ | MINE-Based Measures | | | |
| rm | cb | sg | ds | dn | | mine | mine-loc | mine-glo | mine-glo-corr |
|---|---|---|---|---|---|---|---|---|---|
| | ✓ | ✓ | ✓ | | 2 | 0.470 | 0.292 | 0.337 | **0.278** |
| | ✓ | ✓ | | | 2 | 0.445 | 0.255 | 0.275 | **0.233** |
| | | ✓ | ✓ | | 2 | 1.036 | **0.560** | 0.658 | **0.565** |
| | | ✓ | | | 4 | 1.302 | 0.720 | 0.968 | **0.684** |
| ✓ | | | | ✓ | 4 | 0.930 | 0.438 | 0.803 | **0.381** |
| | | | ✓ | | 4 | 0.276 | **0.369** | 0.642 | 0.375 |
| | | | | ✓ | 4 | 0.622 | 0.423 | 0.895 | **0.269** |
| ✓ | ✓ | | | | 4 | 1.574 | 1.099 | 1.219 | 1.162 |
| | | ✓ | | ✓ | 4 | 1.335 | 0.423 | 0.895 | **0.286** |
| | ✓ | ✓ | | ✓ | 6 | 1.881 | 0.570 | 1.516 | **0.437** |
| | | ✓ | | | 6 | 1.843 | 1.147 | 1.535 | 1.185 |
| | ✓ | | ✓ | | 6 | 1.213 | 0.991 | 1.444 | 1.004 |
| ✓ | | | ✓ | | 6 | 1.014 | 1.088 | 1.441 | 1.058 |
| | | ✓ | | ✓ | 6 | 1.831 | **0.517** | 1.499 | 0.545 |

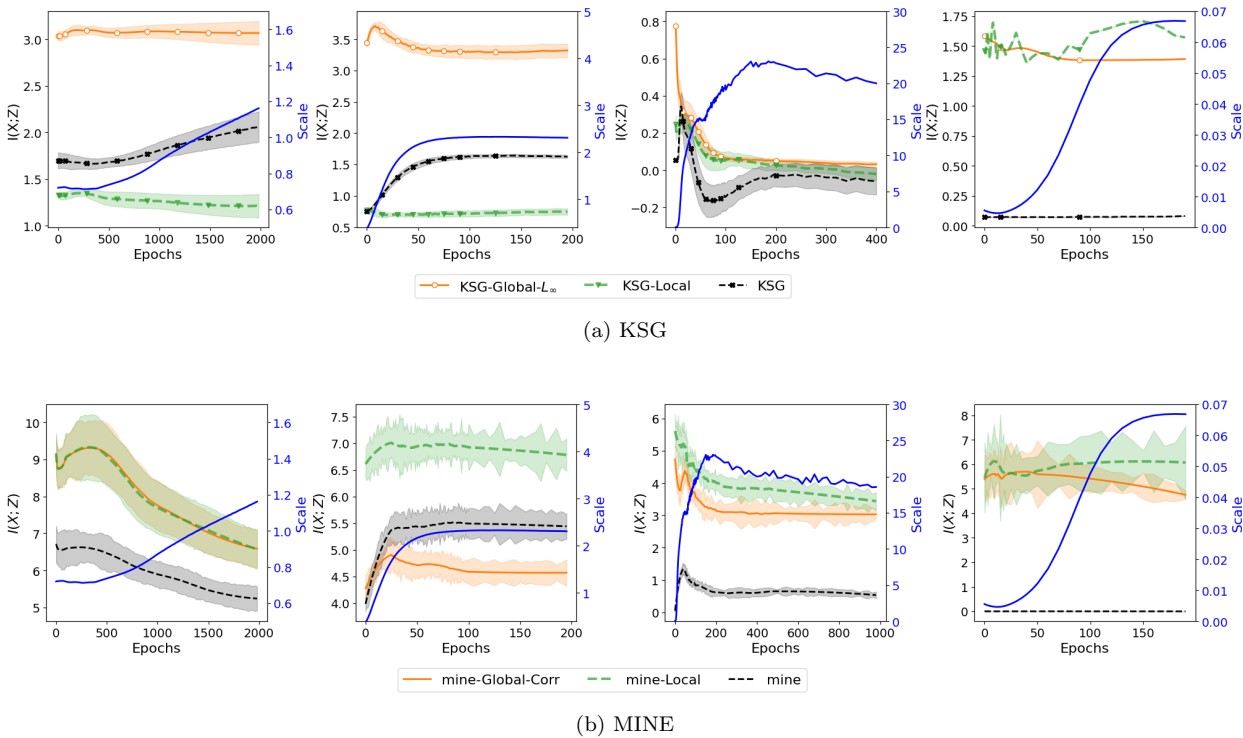

(a) KSG

(b) MINE

Figure 6: (**E7**) $I(X; Z)$ measures estimated after every epoch of training on IB, MNIST, CIFAR-10 and SVHN datasets. $Z$ represents the output of $3^{rd}$ layer for IB dataset and MNIST dataset, and $7^{th}$ layer for CIFAR-10 and SVHN datasets. The scale of the features ($|Z|$) is plotted in blue.

IB and CIFAR-10, it first increases and then decreases (with CIFAR-10 showing a drop after 3 epochs), consistent with the fitting-compression trend of (Shwartz-Ziv & Tishby, 2017).

• **Information Plane Visualization:** We also plot $I(X; Z)$ vs. $I(Z; Y)$ (in Appendix I) to illustrate the trade-off between input representation and label relevance.

Figure 6 reveals distinct behaviors of the proposed MI estimators across datasets:

**IB and MNIST:** On the IB dataset, both local and global variants of KSG and MINE estimators successfully exhibit the fitting (increase in $I(X; Z)$) and compression (decrease in $I(X; Z)$) phases during training, consistent with the information bottleneck theory (Shwartz-Ziv & Tishby, 2017). However, on the MNIST dataset, only the global variants, KSG-Global-$L_\infty$ and MINE-Global-Corrected, clearly demonstrate these trends, while the vanilla variants primarily track the feature scale ($|Z|$) rather than capturing intrinsic mutual information dynamics. This suggests that the normalized variants, which have been tested in the previous section accuracy-wise, can yield different $I(X; Z)$ trends than their vanilla counterparts.

**CIFAR-10 and SVHN:** For CIFAR-10, an interesting trend emerges. Vanilla KSG and MINE estimators show fitting and compression phases, but the fitting phase appears to strongly correlate with initial scale jumps of $|Z|$, indicating sensitivity to feature scale. In contrast, the normalized variants predominantly display compression-like trends throughout training, highlighting a different behavior. Notably, both KSG-Global-$L_\infty$ and MINE-Global-Corr reveal a compression phase from the start.

In SVHN, we see that the unnormalized KSG and MINE both fail to reveal any significant trend and are very close to zero throughout. In contrast both the local and global variants reveal some interesting trends and differences. For both KSG and MINE, global variants suggest that there is no explicit fitting phase (increasing $I(X; Z)$), but rather a slow compression phase throughout. Local variants are less consistent in their trends and they don't capture this slow compression mechanism.

## 5 Conclusion and Limitations

We conducted a comprehensive study on scale invariance in MI estimators, analyzing its impact on estimation accuracy and neural network training. We proposed multiple normalization approaches for KSG and MINE, evaluating their effectiveness in high-dimensional, low-data regimes. Extensive experiments across diverse settings showed that while both local and global normalization have strengths, global variants generally perform better. Finally, MI dynamics during training were analyzed on real datasets, where global normalization variants revealed interesting dynamics different from other normalized and unnormalized approaches. Overall, our work highlights the importance of scale-awareness in the problem of MI estimation, and its potential impact on MI estimates.

**Limitations:** While KSG-Global normalization variants demonstrate superior performance, their computational cost is notably higher, as the MI estimation process requires $m$ times the computation time compared to standard KSG variants. Furthermore, the intrinsic dimensionality of the data remains a fundamental performance bottleneck for these estimators, in spite of normalization, and nonlinear diffeomorphisms such as the cube function generally lead to lower accuracy. We see that while global normalization and its variants remain the better choice in most settings, heavy-tailed distributions do impact performance regardless of choice of normalization. Lastly, while in this work we focus mainly on KSG and MINE, in recent years other estimators have been proposed and popularized, including variants of binning estimators (e.g., Anandkumar et al. (2014b); Vandermeulen & Ledent (2021b)) and other neural network based estimation approaches (e.g., InfoNCE Oord et al. (2018)). We are considering the investigation of normalization methods for these estimators for future work.

## Acknowledgements

This research is supported by A*STAR, CISCO Systems (USA) Pte. Ltd and the National University of Singapore under its Cisco-NUS Accelerated Digital Economy Corporate Laboratory (Award I21001E0002). We also thank the Kent-Ridge AI research group at the National University of Singapore for helpful discussions.

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

# A  Background on Mutual Information Estimators

## A.1  Mutual Information

Mutual information of two variables is a statistical measure that quantifies the mutual dependence between two RVs. Specifically, it measures the amount of information obtained about one random variable through the observation of another. To understand mutual information, it is essential to first examine another foundational concept, Shannon entropy. Shannon entropy represents the intrinsic informational uncertainty associated with a probabilistic system. Given a continuous RV $X$ with a probability density function $f$ from a set $\mathcal{X}$, the continuous entropy $h(X)$ is defined as:

$$h(X) := -\int_{\mathcal{X}} f(x) \log f(x)\, dx. \tag{2}$$

Then, the mutual information between continuous RVs $X$ and $Y$ is given by:

$$I(X;Y) = h(X) + h(Y) - h(X,Y), \tag{3}$$

where $h(X,Y)$ represents the joint differential entropy of $X$ and $Y$, defined as $h(X,Y) = -\int_{\mathcal{X},\mathcal{Y}} f(x,y) \log f(x,y)\, dxdy$. Mutual information can be interpreted as the reduction in the uncertainty of $X$ due to the knowledge of $Y$, or equivalently, as the amount of information that $X$ and $Y$ share.

In the case of jointly continuous RVs, the mutual information can be expressed in terms of Kullback–Leibler (KL-) divergence

$$I(X;Y) = D_{\mathrm{KL}}\left(P(X,Y)\|P(X) \otimes P(Y)\right), \tag{4}$$

where $P(X) \otimes P(Y)$ is the product of two marginal distributions $P(X)$ and $P(Y)$, $P(X,Y)$ is their joint distribution. $D_{\mathrm{KL}}$ is defined as

$$D_{\mathrm{KL}}(P\|Q) := \mathbb{E}_P\left[\log \frac{dP}{dQ}\right]. \tag{5}$$

In practice, estimating the true distribution of continuous RVs is challenging, especially for high-dimensional data. In the following section, we will discuss various non-parametric MI estimators, which estimate the distribution of RVs and subsequently compute estimated mutual information.

## A.2  Mutual Information Estimators

The overall setting of the MI estimation problem is as follows. We are given two RVs $X, T \sim P(X,T)$ and sampled data $S = \{(X_1, T_1), \ldots, (X_n, T_n)\}$. Our estimators are denoted in the form $\widehat{I}_{est}^n(X;T)$, where $est$ denotes the name of the estimator.

In this section, we present several widely-used nonparametric MI estimators that are studied in our work and have been extensively applied in other research.

**Binning Estimator:** Also known as histogram based estimators, this method represents the most direct approach for estimating mutual information. To estimate MI, the continuous random variable is discretized into bins, counting the number of samples that fall into each bin, and computing the probability density (Paninski, 2003). There are many approaches in literature for estimating the bin edges. However, in our theoretical analysis, we consider a simple approach that places a fixed number of bins per dimension, according to the extrema of the individual dimensions.

Generally, the binning estimator for $n$ samples can be expressed as: $\widehat{I}_{bin}^n(X;T) = \widehat{H}_{bin}(X) + \widehat{H}_{bin}(T) - \widehat{H}_{bin}(X,T)$. where $\widehat{H}_{bin}(X)$ represents the binned entropy given a RV $X$, such that $\widehat{H}_{bin}(X) = -\sum_i \widehat{P}(X_i)\, \log \widehat{P}(X_i)$. Let $n(X_i)$ be the number of samples that fall in the $i^{th}$ bin of $X$. Then we have $\widehat{P}(X_i) \approx n(X_i)/n$ for the histogram based estimator. Similarly, we represent binned joint entropy as $\widehat{H}_{bin}(X,T) = -\sum_{i,j} \widehat{P}(X_i, T_j)\, \log \widehat{P}(X_i, T_j)$, and $\widehat{P}(X_i, T_j) \approx n(X_i, T_j)/n$.

While histogram-based estimators are widely acknowledged for their simplicity and lack of scale sensitivity, their vulnerability to the curse of dimensionality limits their applicability in high-dimensional settings. However, recent work in structured density estimation has proposed histogram-like methods that circumvent these limitations by imposing structural assumptions on the underlying density, such as low intrinsic dimension. For instance, low-rank tensor decompositions (Anandkumar et al., 2014a; Vandermeulen & Ledent, 2021a; Amiridi et al., 2022a;b) enable scalable estimation by capturing latent low-dimensional structure, while learnable neural architectures (Vandermeulen et al., 2024) flexibly adapt to complex dependencies. These structured approaches highlight the importance of using inductive biases to lessen the curse of dimensionality when estimating the distribution via histogram-based approaches. Though such models require additional assumptions, they offer promising solutions for histogram-based MI estimation.

**Kraskov–Stögbauer–Grassberger (KSG) Estimator:** Another popular non-parametric approach to estimate MI in high dimensions is the KSG estimator in (Kraskov et al., 2004). Unlike the binning estimator, the KSG estimator uses the $k$-nearest neighbor ($K$-NN) statistic to estimate the probability function of continuous RVs, which also uses the joint entropy decomposition method to estimate MI. The KSG estimator effectively uses the k-nearest neighbor distances to estimate the various entropies involved in the joint-entropy decomposition of $I(X;T)$. We will need some prerequisites to define the KSG estimator. First, let the $k$-NN distance $\rho_{k,i,p}$ be defined as the distance from $(X_i, T_i)$ to $k$-th nearest neighbor in the joint space $(X,T)$ as measured in $l_p$ distance. We now denote $n_{x,i,p} = \sum_{j\neq i} \mathbb{I}\left\{\|X_j - X_i\|_p \leq \rho_{k,i,p}\right\}$ as the number of neighbors of the $i$-th sample $X_i$ within a specified distance under the $l_p$ norm, and similarly for $T$ as $n_{t,i,p}$. Eventually, the KSG estimator yields the following estimate of $I(X;T)$:

$$\widehat{I}^n_{KSG}(X;T) = \psi(k) + \psi(n) - \frac{1}{k} - \frac{1}{n}\sum_{i=1}^n \left(\psi(n_{x,i,\infty}) + \psi(n_{t,i,\infty})\right), \tag{6}$$

where $\psi(x)$ is the digamma function (i.e., $\psi(x) = \Gamma(x)^{-1} d\Gamma(x)/dx$).

In (Gao et al., 2017), authors proposed a bias-improved KSG (**BI-KSG**) that performs better than KSG when $n$ is small and $X$ and $T$ are not independent. It is also important to note that many other variants of KSG and other estimators (Pál et al., 2010; Gao et al., 2015) use a $k$-NN based approach.

**Mutual Information Neural Estimator (MINE):** In our work, we utilize neural network based MI estimators, specifically Mutual Information Neural Estimation (Belghazi et al., 2018). This approach estimates mutual information by using a dual representation of the KL-divergence, known as the Donsker-Varadhan (DV) representation (Donsker & Varadhan, 1983). Given RVs $X \sim P(X)$, $T \sim P(T)$, we express 4 in terms of DV representation as:

$$I(X;T) = \sup_{F:\mathcal{X}\times\mathcal{T}\to\mathbb{R}} \mathbb{E}_{X,T\sim P(X,T)}[F(X,T)] - \log\left(\mathbb{E}_{X,T\sim P(X)\times P(T)}\left[e^{F(X,T)}\right]\right), \tag{7}$$

where $F$ can be any measurable function from $\mathcal{X}\times\mathcal{T}\to\mathbb{R}$ that satisfies the necessary integrability constraints of two expectations in 7 to be well-defined, and the supremum is taken over all such functions.

To compute $I(X;T)$ in practice, first we assume $n$ independent and identically distributed samples (i.i.d.) are drawn from $\{(X_1, T_1), \ldots, (X_n, T_n)\} \sim P(X,T)$. Next, $(X_i, \tilde{T}_i)$, is artificially constructed by choosing $\tilde{T}_i$ as a randomly shuffled set of $(T_i)_{i=1}^n$. When $n$ is large enough, the MINE estimator approximates the MI as:

$$\widehat{I}_{\mathrm{MINE}}(X;T) = \sup_{\theta\in\Theta} \frac{1}{n}\sum_{i=1}^n F_\theta\left(X_i, T_i\right) - \log\left(\frac{1}{n}\sum_{i=1}^n e^{F_\theta\left(X_i, \tilde{T}_i\right)}\right), \tag{8}$$

where $F_\theta : \mathcal{X}\times\mathcal{T}\to\mathbb{R}$ is parameterized by a deep neural network with parameters $\theta\in\Theta$, and $\Theta$ is the parameter space of the neural network. By training a neural network to optimize the above equation (i.e., finding the optimal $\theta^*\in\Theta$), the final output will yield the true MI between $X$ and $T$.

# B  Additional Discussions on Related Works

## B.1  Equitability

There is a body of work on the equitability of mutual information estimators, of which the notion of self-consistent equitability is related to our work. The first work that proposes the notion of self-consistent equitability (Reshef et al., 2013), which is a generalized form of one-sided scale invariance discussed in our work, as we consider the specific case when the function is scale-related. In our work, however, we do not focus on the aspect of data transformation specifically, but mainly focus on the behaviour of estimators in response to scaling. In such works concerned with self-consistent equitability, there is always a very wide range of functions (Table 2 of (Reshef et al., 2013)) to get a broad overview of the behavior of MI in response to any type of transformation. However, it is notable that scaling has not been one of them, most likely due to the way the data was already preprocessed to be scale-invariant, using local normalization, which already makes them self-equitable to scaling.

Our work only indirectly tests self-consistent equitability in Tables 1 and 2, where instead of prioritizing the degree of self-consistent equitability, we prioritize the MI estimation accuracy. We also use a variety of transformations as functions on the data (in a cascaded manner) and see the resulting estimation errors for the various compared metrics. However, our choice of transformations (apart from cube and sigmoid) is very different when compared to Table 2 of Reshef et al. (2013)'s work, as they either encompass all dimensions (like invertible random matrix multiplication) or change the dimensionality of the input by adding noisy dimensions or duplicate dimensions. In contrast, most functions in self-consistent equitability literature work with one-dimensional transformations applied to each data dimension individually. Lastly, as we cascade a random subset of these operations in a random order, we get a richer set of MI-preserving transformations, which can potentially also be tested in an equitability setup using their choice of metrics (like in Figure 2 of (Kinney & Atwal, 2014)). We are considering this for future work.

## B.2  Recent MI Estimation works

We outline three different aspects in which our work differs from (Czyż et al., 2023), which is an important recent study that also tests MI Estimators and their performance under various settings.

**Choice of transformations:** As the focus in (Czyż et al., 2023) is not particularly on the neural network use case where $X$ is the input and $Z$ represents a feature layer, their choice of transformations is motivated differently. Most of their transformations are dimension-wise, i.e., a transformation applied to each dimension. The only exception is their spiral diffeomorphism (Figure 5 of (Czyż et al., 2023)), which radially morphs the distribution in such a way that the MI is preserved. We note that as our focus is mainly on the natural use cases of MI in deep learning, we construct certain types of transformations relevant to this setting. Apart from the cubic transformation, which is dimension-wise, all our other transformations are motivated by the potential use case in deep learning. We outline each one as follows. The sigmoid transformation is motivated by potential uses of sigmoid in the network's hidden layers. The random matrix multiplication (randmat) transformation is motivated via the features undergoing similar transformations through neural network layers, which are usually matrix multiplications followed by non-linearity. Note that the randmat also has an additional scaling term $\alpha$ (Section 4.3), which scales the resulting transformed vector as we wish to also focus on the robustness to scale. The duplicate-noise transformation, which adds dummy noise dimensions, is motivated by the fact that the number of hidden neurons can change through the layers and often many of these dimensions deeper within the network are usually very sparse and noisy. Similarly, the duplicate-self transformation is another approach to changing the dimensionality of the input while preserving the total information.

**Preprocessing methods:** Czyż et al. (2023)'s work indeed finds that the Gaussianization-based local normalization yields better results than uniform marginalization, when the base distribution is the multivariate student distribution. However, they do note that the improvements are minor, and overall they end up choosing the standard variance-based local normalization over the other two. Our conjecture is that Gaussianization may work better when the distribution has long tails, as long-tail distributions are typically harder to estimate MI for, which was observed in (Czyż et al., 2023). This is potentially because the data

samples that are a part of the long tail may end up adding more noise to the final estimate, than in a typical case with Gaussian variables, and Gaussianizing the data preserves MI while avoiding long tails in the input, thereby leading to better performance. A potential direction of future work is thus incorporating similar considerations for our global normalization strategies to further enhance the accuracy of MI estimators.

**Scale invariance:** As the data was already preprocessed using local normalization approaches, scale invariance isn't a part of the analysis in (Czyż et al., 2023), similar to (Kinney & Atwal, 2014). In our case, all our proposed normalization approaches are focused on scale invariance while retaining other desired properties which may not hold for standard normalization approaches, such as robustness to noisy dimensions.

### B.3 Neural network-based pre-processing

As one of our contributions is a set of new pre-processing strategies to ensure desirable scale-invariant behaviour of MI estimators, we discuss how this compares to other recent work that uses neural networks to learn pre-processing strategies for more accurate MI Estimation.

Gowri et al. (2024): In this work, the authors propose a two-step MI estimation procedure. First, they train compressed representations of the input which minimize an upper bound on the conditional entropies between the compressed representations and the input/output, after which they estimate the MI between the compressed representations using KSG. In the limiting case, these compressed representations would preserve the true MI, otherwise, normally they are upper-bounded by it. It is worth noting their compressed representations can be of any isomorphic form w.r.t the choice of compressor, and as they use neural networks, the scale of the compression can vary. So to alleviate the issue, they use the standard unit normalization approach (local). As such, our findings in this work, including the different variants proposed for global normalization, can be applied to their KSG estimation step to generate more robust MI estimates.

Butakov et al. (2024): It seemed to us that this work proposed a new estimator using normalizing flows, which in this case are function maps (which can be a neural network) which eventually transform the data distribution to a simpler one which has a closed form solution for MI estimation. In their work, they seem to focus on a Gaussian base distribution. In principle, their estimator could be scale invariant, as their final estimator is the analytical MI expression itself. However, we did not see any explicit theoretical study on this, as one will need to show that the correlation matrix at the end of applying multiple normalizing flows is invariant to one-sided scaling of one of the variables. Overall, our work points out that scale-invariance is an important consideration for any MI estimator to prevent scale confounding in the estimates.

### B.4 Contrastive Learning-based MI Estimators

Contrastive learning-based MI estimators, such as InfoNCE (Oord et al., 2018), estimate mutual information by maximizing the similarity between positive pairs while minimizing similarity between negative pairs. However, we can show that these estimators do not satisfy **one-sided scale invariance**, which is a key focus of our work. InfoNCE estimates MI by optimizing the following loss function:

$$\mathcal{L}_{\text{InfoNCE}}(X, Z) = -\mathbb{E}\left[\log \frac{e^{f(X,Z)}}{\sum_{j=1}^{n} e^{f(X,Z_j)}}\right],$$

where $f(X, Z)$ is a learned similarity function, often a dot product or cosine similarity. The true MI is computed as the lower bound $I(X; Z) \geq \log n - \mathcal{L}_{\text{InfoNCE}}(X, Z)$. In the original work, they used a dot-product like function, yielding $f(X, Z) = g(X)^T W Z$. We note that when $Z$ is scaled here, we obtain $f(X, \alpha Z) = \alpha f(X, Z)$. However this yields $\mathcal{L}_{\text{InfoNCE}}(X, \alpha Z) = -\mathbb{E}\left[\log \frac{e^{\alpha f(X,Z)}}{\sum_j e^{\alpha f(X,Z_j)}}\right]$. This will not be scale invariant, as in the limiting case when $\alpha \to 0^+$, we can see that $\log \frac{e^{\alpha f(X,Z)}}{\sum_j e^{\alpha f(X,Z_j)}} = \log \frac{1}{n}$, thus $\lim_{\alpha \to 0} \mathcal{L}_{\text{InfoNCE}}(X, \alpha Z) = \log n$ in this scenario. Thus, the lower bound in this case would be $I(X; Z) \geq \log n - \mathcal{L}_{\text{InfoNCE}}(X, Z) = 0$. Having noted that, applying local or global normalization to $X$ and $Z$ should similarly yield a scale invariant estimator, so it is worthwhile exploring our approaches in this setting for future work.

# C   Appendix: Details for Experiments

## C.1   Details for the Section 5: Experimental Studies

**Key Parameters:**

**Figure 1: Average MI estimates for KSG for a varying number of noise dimensions**

- **Setup:** Additive Gaussian ($X, T \in \mathbb{R}^2$ where $T = X + \epsilon$, with $\epsilon \sim \mathcal{N}\left(0, \sigma^2 I_2\right)$)
- **Number of Samples:** 1000
- **Number of Trials:** 10
- $\sigma = \frac{1}{\sqrt{2}}$
- $\sigma' = 0.04$

**Figure 2: Bias of MINE-based measures for varying noise dimensions**

- **Setup:** Correlated Gaussian $X, T \in \mathbb{R}^2$ with correlated coefficient $\rho$
- **Number of Samples**: 1000
- **Number of Trials:** 10
- **Correlation coefficient**: $\rho$: 0.2

**Figure 3: Bias of the KSG estimator with the real data dimension**

**Figure 3(a):**

- **Setup:** Correlated Gaussian $X, T \in \mathbb{R}^d$ with correlated coefficient $\rho$
- **Number of Samples**: 1000
- **Number of Trials:** 20
- **Dimensionality ($d$):** Evaluated over $d \in \{1, 2, 3, 4, 5, 6, 7, 8, 9\}$

**Figure 3(b):**

- **Setup:** Correlated Gaussian $X, T \in \mathbb{R}^d$ with correlated coefficient $\rho$
- **Number of Samples**: 200
- **Number of Trials:** 20
- **Correlation coefficient**: $\rho \sim Unif[0.4, 0.9]$, where $Unif$ denotes the uniform distribution
- **Dimensionality ($d$):** Evaluated over $d \in \{2, 4, 8, 16, 32, 64\}$, and $\log_2 d \in \{1, 2, 3, 4, 5, 6\}$.

**Figure 4: Data duplication: Average MI estimates for KSG-based approaches.**

- **Setup:** Additive Gaussian ($X, T \in \mathbb{R}^2$ where $T = X + \epsilon$, with $\epsilon \sim \mathcal{N}\left(0, \sigma^2 I_2\right)$), and $X' = [X, X, X, \ldots, X]$.
- **Number of Samples:** 1000
- **Number of Trials:** 10
- $\sigma = \frac{1}{\sqrt{2}}$
- **Dimensionality of Final Input:** $d_{X'} \in \{2, 22, 42, 62, 82, 102, 122, 142, 162, 182\}$

**Figure 5: Estimator bias versus dimension: Comparing MINE with MINE-Global variants**

- **Setup:** Correlated Gaussian $X, T \in \mathbb{R}^d$ with correlated coefficient $\rho$
- **Number of Samples**: 1000
- **Number of Trials:** 10
- **Correlation coefficient**: $\rho$: 0.2
- **Dimensionality ($d$):** Evaluated over $d_X, d_T \in \{1, 2, 3, 4, 5, 6, 7, 8, 9\}$

**Figure 6: Analysis of MI Estimators in response to data scaling. Estimates are for $I(\eta X; T)$, where $\eta$ is the scaling factor**

**Figure 6(a) & 6(b):**

- **Setup:** Correlated Gaussian $X, T \in \mathbb{R}^2$ with correlated coefficient $\rho$, estimates are for $I(\eta X; T)$, where $\eta$ is the scaling factor.
- **Number of Samples:** 1000
- **Number of Trials:** 20
- **Correlation Coefficient:** $\rho$: 0.8 (KSG); 0.5 (MINE)
- **Scaling Factor ($\eta$):**
  - For KSG: $\eta \in [10^{-2}, 10^3]$, equispaced on a $\log_{10}$ scale.
  - For MINE: $\eta \in [10^{-2}, 10]$, equispaced on a $\log_{10}$ scale.

**Figure 6(c) & 6(d):**

- **Setup:** Correlated Gaussian $X, T \in \mathbb{R}^2$ with correlated coefficient $\rho$, RMSE of MI estimates $I(\eta X; T)$, where $\eta$ is the scaling factor.
- **Number of Samples:** 1000
- **Number of Trials:** 20
- **Correlation Coefficient:** $\rho \sim Unif[0, 0.8]$
- **Scaling Factor ($\eta$):**
  - For KSG: $\eta \in [10^{-2}, 10^3]$, equispaced on a $\log_{10}$ scale.
  - For MINE: $\eta \in [10^{-2}, 10]$, equispaced on a $\log_{10}$ scale.
- **Error Metric:** RMSE computed between MI estimates and ground truth.

**Figure 7: Average MI estimates for various estimators across different values of SNR.**

- **Setup:** Additive Gaussian noise base, where $T = X + \epsilon$ with $\epsilon \sim \mathcal{N}(0, \sigma^2 I_2)$. Additionally, $T$ is scaled to $T' = 0.1T$, and the mutual information is estimated as $I(X; T')$.
- **Number of Samples:** 1000
- **Number of Trials:** 10
- $\sigma = \frac{1}{\sqrt{SNR}}$
- **Dimensionality ($d$):** $d_X, d_T = 2$

## D  Background: Multivariant Student's t-distribution

**Sampling Process**   To generate correlated samples following the multivariate Student's t-distribution, we construct a dataset where each sample consists of two correlated random variables, $X$ and $T$, with a specified dimensionality and degrees of freedom ($\nu$). The multivariate Student's t-distribution is obtained by first sampling an $(m + n)$-dimensional random vector $(\tilde{X}, \tilde{T}) \sim \mathcal{N}(0, \Omega)$ and a random scalar $U \sim \chi_\nu^2$, where $\Omega$ is a positive definite dispersion matrix. The rescaled variables are then defined as:

$$X = \tilde{X}\sqrt{\frac{\nu}{U}}, \quad T = \tilde{T}\sqrt{\frac{\nu}{U}},$$

which follow the multivariate Student's t-distribution. The tail behavior is controlled by the degrees of freedom $\nu$, where lower values result in heavier tails. Specifically, for $\nu = 1$, the distribution reduces to the multivariate Cauchy distribution, for $\nu = 2$, the mean exists but the covariance does not, and for $\nu > 2$, both the mean and covariance exist, with $\text{Cov}(X, T) = \frac{\nu}{\nu-2}\Omega$. When $\nu \gg 1$, due to the concentration of measure phenomenon, $U$ has most of its probability mass around $\nu$, and the variables $(X, T)$ can be well approximated by $(\tilde{X}, \tilde{T})$.

**Ground Truth Mutual Information Estimation**   The mutual information $I(X; T)$ quantifies the dependency between $X$ and $T$ and can be computed as the sum of the mutual information of the Gaussian-distributed basis variables and a correction term. (Arellano-Valle & Azzalini, 2013) proved that:

$$I(X; T) = I(\tilde{X}; \tilde{T}) + c(\nu, m, n),$$

where the correction term $c(\nu, m, n)$ is given by:

$$c(\nu, m, n) = f(\nu) + f(\nu + m + n) - f(\nu + m) - f(\nu + n),$$

with

$$f(x) = \log \Gamma\left(\frac{x}{2}\right) - \frac{x}{2}\psi\left(\frac{x}{2}\right),$$

where $\psi(x)$ is the digamma function. Unlike the Gaussian case, even when $\Omega = I_{m+n}$, the mutual information $I(X;T) = c(\nu, m, n)$ remains non-zero, as $U$ provides additional information about the magnitude. In our benchmark, we use this dispersion matrix to evaluate how well estimators capture the information contained in the tails, rather than focusing solely on estimating the Gaussian term.

## E  Sampling details for Synthetic Experiments

For our synthetic experiments in Tables 2 and 3 in the main paper, we describe how the various parameters in choosing the distribution base were set (from Section 4).

1. **Correlated Gaussians**: For each trial, we sampled $\rho \sim \mathcal{U}(0, 0.8)$, where $\mathcal{U}(a, b)$ denotes the uniform distribution between $a$ and $b$.

2. **Additive Gaussian Noise**: First let $\sigma = \sqrt{\frac{1}{\alpha}}$. Then for each trial, we sampled $\alpha \sim \mathcal{U}(0, 2.0)$.

3. **Student's t-distribution:** In each trial, a random dof $\nu$ from $\{1, 2, 3, 5, 8\}$ was chosen (uniformly distributed), which is the same set of dofs used in (Czyż et al., 2023), and the dispersion was set as

$$\Omega = \begin{bmatrix} I_{\dim} & \rho I_{\dim} \\ \rho I_{\dim} & I_{\dim} \end{bmatrix},$$

where $\rho$ is sampled as $\rho \sim \mathcal{U}(0, 0.8)$.

## F  MI Estimators: Configurations

### F.1  KSG

We used the NPEET MI estimator toolbox for estimating KSG and KSG-based measures [3]. We set $k = 3$ for all experiments. For the global KSG variants, we fix $c_1 = 0.1, c_2 = 0.2, \ldots, c_m = 2$ for all our experiments.

### F.2  MINE

We used the popular pytorch-based package [4] for the MINE implementation.

**Overall MINE implementation:** For estimating $I(X;T)$, we used single-hidden layer ReLU-activated neural networks of the configuration: $(d_X + d_T) \to H_1 \to H_2 \to \ldots \to H_k \to 1$, where $d_X + d_T$ is the dimensionality of the input and $H_1, \ldots, H_k$ is the number of hidden neurons for each hidden layer. The last layer is a linear layer. We used the Adam optimizer with a learning rate of 0.001. The hidden neuron configuration varies depending on our experiment. We set the number of epochs to 50 for all experiments. Given a training dataset $S = \{(X_1, T_1), \ldots, (X_n, T_n)\}$, the network $F_\theta$ effectively minimizes the following loss:

$$-\frac{1}{n}\sum_{i=1}^{n} F_\theta\left(X_i, T_i\right) + \log\left(\frac{1}{n}\sum_{i=1}^{n} e^{F_\theta\left(X_i, \tilde{T}_i\right)}\right). \tag{9}$$

Note that we estimated MINE in the standard manner as per its original definition, which is, once the networks are optimized, we estimate the above loss on the training dataset Song & Ermon (2019). It is noteworthy that to avoid overestimation, (Czyż et al., 2023) proposes an approach where the MINE estimate is computed on the test data. However, in our case, we find that MINE does not overestimate for most of the cases in Tables 3 and 5, as it almost always has a negative bias, and thus the training/test split approach is not necessary.

---

[3] https://github.com/gregversteeg/NPEET   [4] https://github.com/gtegner/mine-pytorch

**Figures 1b,3b,4,5b and Table 3:** For the small dataset cases, we found that using a smaller number of hidden neurons yielded significantly better and more stable results. Thus, for Figures 1b,3b,4,5b and Table 3, using a single hidden layer with $H_1 = 20$ yielded the most stable results on average, and thus we set $H_1 = 20$. We found that for this small sample size setting, increasing $H_1$ led to unstable estimates and large variance of estimators.

**Figure 6:** For the larger dataset cases, which is the case for our real datasets, we were able to increase $H$, and the details are as follows. For the IB dataset, we have one hidden layer with $H = 30$ neurons. For MNIST and CIFAR-10 datasets, we have two hidden layers with $H = 30$ neurons each. For the SVHN dataset, we have one hidden layer with $H = 100$ neurons. The network layers are ReLU activated.

### F.3 Network Architecture for Neural Network Analysis in Section 6

Table 4: Model Architecture for IB Dataset

| Layer | Dimension | Activation Function |
|---|---|---|
| Input | $28 \times 28$ | - |
| Flatten | 12 | - |
| Dense | 10 | ReLU |
| **Dense** | **7** | **ReLU** |
| Dense | 5 | ReLU |
| Dense | 4 | ReLU |
| Dense | 4 | ReLU |
| Dense | 2 | SoftMax |

Table 5: Model Architecture for MNIST Dataset

| Layer | Dimension | Activation Function |
|---|---|---|
| Input | $28 \times 28$ | - |
| Flatten | 784 | - |
| Dense | 1024 | ReLU |
| **Dense** | **20** | **ReLU** |
| Dense | 20 | ReLU |
| Dense | 20 | ReLU |
| Dense | 10 | SoftMax |

Table 6: Model Architecture for CIFAR-10 Dataset

| Layer | Dimension | Activation Function |
|---|---|---|
| Input | $32 \times 32 \times 3$ | - |
| Conv2D | $32 \times 32 \times 16$ | ReLU |
| Conv2D | $32 \times 32 \times 16$ | ReLU |
| MaxPooling | $16 \times 16 \times 16$ | - |
| Conv2D | $16 \times 16 \times 32$ | ReLU |
| Conv2D | $16 \times 16 \times 32$ | ReLU |
| **Global AveragePooling** | **32** | **-** |
| Dense | 64 | ReLU |
| Dense | 10 | SoftMax |

For the MNIST and IB datasets, we replicate the network architectures from Saxe et al. (2018)'s work, using the widely-adopted *ReLU* activation function for the hidden layers. Specifically, for the IB dataset, we utilize

Table 7: Model Architecture for SVHN Dataset

| Layer | Dimension | Activation Function |
|---|---|---|
| Input | $32 \times 32 \times 3$ | - |
| Conv2D | $32 \times 32 \times 16$ | ReLU |
| Conv2D | $32 \times 32 \times 16$ | ReLU |
| MaxPooling | $16 \times 16 \times 16$ | - |
| Conv2D | $16 \times 16 \times 32$ | ReLU |
| Conv2D | $16 \times 16 \times 32$ | ReLU |
| **Global AveragePooling** | **32** | **-** |
| Dense | 256 | ReLU |
| Dense | 10 | SoftMax |

a neural network with 7 hidden layers of dimensions 12-10-7-5-4-3-2. For the MNIST dataset, the neural network consists of 6 fully connected layers with dimensions 784-1024-20-20-20-10.

For the CIFAR-10 dataset, we adopt a neural network with 4 convolutional layers, 3 fully connected layers. The networks are trained using SGD and cross-entropy loss. We train 2000 epochs for the IB dataset, 200 epochs for the MNIST dataset, and 1000 epochs for the CIFAR-10 dataset.

In Tables 4-7, we present the network architecture and output dimensions for each layer of the neural networks used in our study. The layers with bold text are the layers for extracted $Z$.

For the IB dataset, we trained for 2000 epochs with an SGD optimizer and a learning rate of $5 \times 10^{-3}$. For the MNIST dataset, we trained for 200 epochs with an SGD optimizer and a learning rate of $5 \times 10^{-4}$. For the CIFAR-10 dataset, we trained for 1000 epochs with an SGD optimizer and a learning rate of $1 \times 10^{-3}$. For SVHN, we trained for 200 epochs, with an initial learning rate of $1 \times 10^{-3}$. The batch sizes were 256 for the IB dataset, 128 for the MNIST dataset, and 512 for the CIFAR-10 and SVHN datasets.

# G   Full Results on Synthetic Data

We provide the full results of Tables 2 and 3 of the main paper, in Table 9, and include the full results for the correlated Gaussian distribution and the Student's t-distribution bases in Tables 10 and 8. In the following tables, in addition to the normalized root mean squared error, we also report the Spearman correlation and Bias of each estimator, as defined in Section 4.3.1. Table 9 addresses the full KSG and MINE results in the same setting as Tables 2 and 3, and similarly, tables 10 and 8 address the full KSG and MINE results for the correlated Gaussian distribution and Student's t-distribution bases.

These are the specific takeaways from the additional results that follow:

1. Very interestingly, we find that although the prediction error in terms of normalized RMSE quickly becomes higher than random guessing (red entries) for data in higher dimensions, the Spearman correlation of MI estimates with the ground truth MI still stays high in many cases.

2. A particularly notable example of this is for KSG variants, where we find that in spite of normalized RMSE exceeding 1, the KSG variants still have Spearman correlation measures very close to 1, showing that they are very highly rank-correlated with the true MI. It is therefore clear that the large negative bias of KSG variants in high dimensions affects all cases similarly, and thus the dependency between the estimated MI and the true MI remains. This also highlights that using some carefully calibrated ways of estimating MI in high dimensional settings may yield very accurate predictions in terms of normalized RMSE as well.

3. In fact, we find that over all cases in correlated and additive Gaussian setups, only the KSG-global variant stays very consistent in terms of high Spearman correlation. Barring only two cases, we see that the Spearman correlation of KSG-global-$L_\infty$ is always greater than 0.9.

4. We find that overall our normalized variants of KSG and MINE showcase bias closest to 0. We see that for all variants, the bias roughly becomes increasingly negative as the dimensionality of the input increases. This mirrors our observation in Section 4.1.2.

5. We find that the overall results suffer in the heavy-tailed case of the Student's t-distribution base, especially MINE variants. However, we still see KSG-global-$L_\infty$ show consistently best performance among the KSG variants in most scenarios, and overall find that MINE-glo-corr shows better performance than other MINE variants, although the improvements are less significant in that case.

6. We find that certain transformations are more difficult to handle than others. Notably the cube transformation which non-linearly maps each individual dimension, seems to significantly impact performance negatively. This shows that invariance to diffeomorphisms such as the cube transformation is still a challenging problem.

Table 8: Comparing performance measures of MI Estimators: Correlated Student's t-distribution Base

| Transformation | | | | | d | measure | KSG-Based Measures | | | | | MINE-Based Measures | | | |
|---|---|---|---|---|---|---|---|---|---|---|---|---|---|---|---|
| rm | cb | sg | ds | dn | | | ksg | ksg-loc | ksg-glo | ksg-glo-$L_\infty$ | ksg-revised | mine | mine-loc | mine-glo | mine-glo-corr |
| | ✓ | ✓ | ✓ | | 2 | RMSE-norm | 0.451 | 0.320 | 0.307 | **0.274** | 0.496 | 13.837 | 0.552 | 0.564 | **0.539** |
| | | | | | | spearman | 0.827 | 0.880 | **0.907** | 0.895 | 0.810 | -0.544 | -0.013 | -0.005 | -0.023 |
| | | | | | | bias | -0.716 | -0.471 | -0.484 | **-0.395** | -0.803 | -11.578 | -0.975 | -0.985 | **-0.945** |
| | | ✓ | ✓ | | 2 | RMSE-norm | 0.403 | 0.334 | 0.246 | **0.227** | 0.438 | 13.838 | 0.559 | 0.562 | **0.541** |
| | | | | | | spearman | 0.852 | 0.837 | **0.899** | 0.879 | 0.826 | -0.542 | -0.078 | -0.031 | -0.002 |
| | | | | | | bias | -0.627 | -0.425 | -0.371 | **-0.288** | -0.692 | -11.579 | -0.989 | -0.97 | **-0.957** |
| | ✓ | ✓ | | | 2 | RMSE-norm | 0.475 | 0.340 | **0.287** | 0.286 | 0.524 | 16.314 | **0.568** | 0.582 | 0.743 |
| | | | | | | spearman | 0.814 | **0.863** | **0.870** | 0.868 | 0.783 | -0.519 | 0.098 | 0.009 | -0.125 |
| | | | | | | bias | -0.752 | -0.502 | **-0.418** | -0.419 | -0.849 | -11.864 | -1.052 | -1.074 | -1.236 |
| | | ✓ | | | 4 | RMSE-norm | 0.930 | 0.848 | 0.694 | **0.542** | 1.029 | 10.401 | **0.912** | 0.937 | **0.914** |
| | | | | | | spearman | **0.975** | 0.209 | 0.639 | 0.883 | **0.969** | -0.429 | -0.348 | -0.277 | -0.31 |
| | | | | | | bias | -0.974 | -0.795 | -0.649 | **-0.552** | -1.126 | -6.248 | -1.406 | -1.506 | -1.416 |
| ✓ | | ✓ | ✓ | ✓ | 4 | RMSE-norm | 0.685 | 0.594 | 0.581 | **0.460** | 0.739 | 18.908 | **0.831** | 0.881 | 0.843 |
| | | | | | | spearman | 0.789 | 0.898 | 0.947 | **0.959** | 0.737 | -0.309 | -0.153 | -0.112 | -0.248 |
| | | | | | | bias | -0.704 | -0.596 | -0.614 | **-0.489** | -0.770 | -12.302 | -1.119 | -1.371 | -1.18 |
| | | | ✓ | | 4 | RMSE-norm | **0.952** | 0.472 | 0.581 | **0.395** | 0.406 | 19.416 | 0.804 | 0.857 | **0.756** |
| | | | | | | spearman | **0.952** | 0.935 | **0.961** | **0.961** | **0.955** | -0.371 | -0.146 | -0.152 | 0.101 |
| | | | | | | bias | -0.437 | -0.485 | -0.610 | **-0.420** | -0.443 | -12.627 | -1.05 | -1.27 | -1.043 |
| | | | | ✓ | 4 | RMSE-norm | 0.403 | 1.136 | **0.393** | 0.395 | 0.406 | 12.782 | 1.033 | **0.89** | **0.892** |
| | | | | | | spearman | **0.952** | 0.050 | **0.961** | **0.961** | **0.955** | -0.347 | -0.558 | -0.271 | -0.419 |
| | | | | | | bias | -0.437 | -1.230 | -0.418 | **-0.420** | -0.443 | -6.84 | 0.209 | -1.347 | -0.576 |
| ✓ | ✓ | | | | 4 | RMSE-norm | 1.009 | 1.133 | 1.125 | 1.088 | 1.094 | 149915.3 | **0.986** | 1.029 | 1.023 |
| | | | | | | spearman | **0.666** | -0.270 | -0.232 | -0.067 | 0.438 | -0.103 | 0.07 | -0.38 | -0.247 |
| | | | | | | bias | -1.081 | -1.217 | -1.203 | -1.157 | -1.210 | -28498.3 | -1.676 | -1.744 | -1.716 |
| | ✓ | ✓ | ✓ | ✓ | 4 | RMSE-norm | 0.930 | 1.153 | 0.691 | **0.542** | 1.029 | 11.404 | 1.038 | **0.889** | 0.897 |
| | | | | | | spearman | **0.975** | -0.090 | 0.643 | 0.883 | **0.969** | -0.388 | -0.52 | -0.381 | -0.446 |
| | | | | | | bias | -0.974 | -1.254 | -0.648 | **-0.552** | -1.126 | -5.985 | 0.173 | -1.321 | -0.515 |
| | ✓ | ✓ | | ✓ | 6 | RMSE-norm | 1.121 | 1.200 | 0.861 | **0.766** | 1.252 | 11.812 | 1.498 | 1.086 | **1.099** |
| | | | | | | spearman | 0.727 | -0.221 | 0.535 | **0.899** | 0.675 | -0.095 | -0.5 | -0.309 | -0.319 |
| | | | | | | bias | -1.199 | -1.303 | -0.877 | **-0.800** | -1.404 | -6.397 | 0.759 | -1.713 | **-0.324** |
| | | ✓ | | | 6 | RMSE-norm | **1.148** | 0.988 | 0.842 | **0.703** | 1.282 | 13.756 | 1.096 | 1.13 | **1.08** |
| | | | | | | spearman | 0.750 | -0.038 | 0.377 | **0.807** | 0.693 | -0.192 | -0.233 | -0.261 | -0.269 |
| | | | | | | bias | -1.238 | -0.960 | -0.810 | **-0.697** | -1.446 | -8.374 | -1.712 | -1.859 | -1.694 |
| | ✓ | | ✓ | | 6 | RMSE-norm | 0.933 | 1.163 | 1.201 | 1.125 | 0.939 | 6964826 | 1.456 | 1.164 | **1.2** |
| | | | | | | spearman | **0.962** | -0.423 | -0.430 | -0.335 | **0.969** | 0.142 | -0.357 | -0.182 | -0.22 |
| | | | | | | bias | **-0.973** | -1.244 | -1.297 | -1.184 | -0.993 | 1937815 | -2.042 | -1.959 | -1.958 |
| ✓ | | | ✓ | | 6 | RMSE-norm | 0.865 | 0.801 | 0.942 | **0.725** | 0.930 | 15.315 | 1.054 | 1.114 | **1.034** |
| | | | | | | spearman | 0.809 | 0.958 | 0.926 | **0.970** | 0.735 | -0.165 | -0.084 | -0.204 | 0.094 |
| | | | | | | bias | -0.942 | -0.844 | -0.993 | **-0.759** | -1.040 | -10.795 | -1.644 | -1.832 | -1.636 |
| | | ✓ | | ✓ | 6 | RMSE-norm | **1.148** | 1.209 | 0.837 | **0.716** | 1.282 | 11.816 | 1.534 | 1.034 | **1.597** |
| | | | | | | spearman | 0.750 | -0.375 | 0.412 | **0.797** | 0.693 | -0.092 | -0.484 | -0.38 | -0.394 |
| | | | | | | bias | -1.238 | -1.316 | -0.808 | **-0.709** | -1.446 | -6.398 | 0.809 | -1.466 | 0.898 |

Table 9: Comparing performance measures of MI Estimators: Additive Gaussian Noise Base (Full Results)

| rm | cb | sg | ds | dn | d | measure | ksg | bi-ksg | ksg-loc | ksg-glo | ksg-glo-$L_\infty$ | mine | mine-loc | mine-glo | mine-glo-corr |
|---|---|---|---|---|---|---|---|---|---|---|---|---|---|---|---|
| | ✓ | ✓ | ✓ | ✓ | 2 | RMSE-norm | 0.351 | 0.404 | 0.147 | 0.208 | 0.110 | 0.470 | 0.292 | 0.337 | **0.278** |
| | | | | | | spearman | 0.985 | 0.985 | 0.983 | 0.982 | 0.983 | **0.938** | 0.920 | 0.889 | 0.928 |
| | | | | | | bias | -0.191 | -0.226 | -0.072 | -0.105 | **-0.047** | -0.246 | -0.130 | -0.161 | **-0.117** |
| | | ✓ | ✓ | | 2 | RMSE-norm | 0.286 | 0.335 | 0.060 | 0.084 | **0.050** | 0.445 | 0.255 | 0.275 | **0.233** |
| | | | | | | spearman | 0.992 | 0.992 | 0.988 | 0.989 | 0.990 | **0.938** | **0.929** | 0.917 | **0.932** |
| | | | | | | bias | -0.160 | -0.192 | -0.013 | -0.035 | **0.004** | -0.233 | -0.108 | -0.126 | **-0.093** |
| | ✓ | | ✓ | | 2 | RMSE-norm | 0.458 | 0.533 | 0.145 | **0.113** | 0.113 | 1.036 | **0.560** | 0.658 | **0.565** |
| | | | | | | spearman | 0.982 | 0.981 | 0.986 | 0.987 | 0.987 | 0.548 | 0.816 | 0.853 | **0.884** |
| | | | | | | bias | -0.253 | -0.302 | -0.071 | **-0.050** | -0.050 | -0.559 | **-0.283** | -0.340 | -0.292 |
| | | ✓ | | | 4 | RMSE-norm | 1.275 | 1.424 | 0.312 | **0.300** | 0.301 | 1.302 | 0.720 | 0.968 | **0.684** |
| | | | | | | spearman | 0.988 | 0.991 | 0.993 | 0.995 | 0.995 | 0.805 | **0.930** | 0.901 | **0.922** |
| | | | | | | bias | -1.078 | -1.235 | -0.254 | **-0.240** | -0.240 | -1.105 | -0.566 | -0.781 | **-0.543** |
| ✓ | | | | ✓ | 4 | RMSE-norm | 0.862 | 0.932 | 0.445 | 0.594 | **0.396** | 0.930 | 0.438 | 0.803 | **0.381** |
| | | | | | | spearman | 0.599 | 0.549 | **0.995** | 0.990 | 0.994 | 0.660 | 0.897 | 0.944 | **0.965** |
| | | | | | | bias | -0.653 | -0.714 | -0.363 | -0.490 | **-0.322** | -0.738 | -0.296 | -0.640 | **-0.265** |
| | | | ✓ | | 4 | RMSE-norm | 0.332 | 0.342 | **0.304** | 0.520 | 0.297 | 0.276 | **0.369** | 0.642 | 0.375 |
| | | | | | | spearman | 0.994 | 0.994 | 0.995 | 0.994 | 0.994 | 0.931 | 0.922 | 0.922 | **0.951** |
| | | | | | | bias | -0.282 | -0.298 | -0.247 | -0.426 | **-0.239** | -0.122 | -0.223 | -0.494 | **-0.232** |
| | | | | ✓ | 4 | RMSE-norm | 0.332 | 0.342 | 1.327 | **0.298** | 0.297 | 0.622 | 0.423 | 0.895 | **0.269** |
| | | | | | | spearman | **0.994** | **0.994** | 0.954 | **0.995** | 0.994 | 0.876 | 0.865 | 0.917 | **0.942** |
| | | | | | | bias | -0.282 | -0.298 | -1.107 | **-0.239** | -0.239 | -0.486 | **0.002** | -0.719 | 0.118 |
| ✓ | ✓ | | | | 4 | RMSE-norm | 1.131 | 1.233 | 0.977 | 0.959 | **0.931** | 1.574 | 1.099 | 1.219 | 1.162 |
| | | | | | | spearman | 0.588 | 0.534 | **0.975** | 0.961 | 0.969 | 0.542 | **0.905** | 0.837 | 0.769 |
| | | | | | | bias | -0.942 | -1.045 | -0.816 | -0.796 | **-0.773** | -1.198 | **-0.914** | -1.012 | -0.960 |
| | ✓ | ✓ | | | 4 | RMSE-norm | 1.275 | 1.424 | 1.334 | **0.300** | 0.301 | 1.335 | 0.423 | 0.895 | **0.286** |
| | | | | | | spearman | **0.988** | **0.991** | 0.916 | **0.995** | 0.995 | 0.807 | 0.870 | 0.914 | **0.939** |
| | | | | | | bias | -1.078 | -1.235 | -1.113 | **-0.239** | -0.240 | -1.119 | **-0.004** | -0.718 | 0.148 |
| | ✓ | ✓ | | ✓ | 6 | RMSE-norm | 1.981 | 2.129 | 1.904 | 1.021 | 1.021 | 1.881 | 0.570 | 1.516 | **0.437** |
| | | | | | | spearman | 0.969 | 0.966 | 0.905 | **0.989** | 0.989 | 0.852 | 0.786 | **0.881** | 0.855 |
| | | | | | | bias | -1.999 | -2.174 | -1.919 | -1.021 | -1.021 | -1.898 | **-0.179** | -1.502 | **-0.181** |
| | | ✓ | | | 6 | RMSE-norm | 1.983 | 2.131 | **0.816** | 0.811 | 0.812 | 1.843 | 1.147 | 1.535 | 1.185 |
| | | | | | | spearman | 0.959 | 0.957 | **0.995** | 0.995 | 0.995 | 0.842 | 0.771 | **0.862** | 0.815 |
| | | | | | | bias | -2.001 | -2.175 | **-0.816** | -0.809 | -0.810 | -1.861 | -1.116 | -1.527 | -1.155 |
| ✓ | | | ✓ | | 6 | RMSE-norm | 1.377 | 1.408 | 1.343 | 1.605 | **1.290** | 1.213 | **0.991** | 1.444 | 1.004 |
| | | | | | | spearman | **0.983** | **0.981** | 0.979 | 0.968 | 0.989 | 0.694 | 0.783 | 0.831 | **0.881** |
| | | | | | | bias | -1.377 | -1.410 | -1.347 | -1.609 | **-1.291** | -1.158 | **-0.933** | -1.432 | -0.971 |
| | | | ✓ | | 6 | RMSE-norm | 1.643 | 1.730 | 1.275 | 1.660 | **1.153** | 1.014 | 1.088 | 1.441 | 1.058 |
| | | | | | | spearman | 0.338 | 0.314 | 0.937 | 0.871 | **0.976** | 0.775 | 0.798 | **0.841** | 0.818 |
| | | | | | | bias | -1.631 | -1.730 | -1.275 | -1.666 | **-1.150** | -0.967 | -1.042 | -1.431 | -1.028 |
| ✓ | | ✓ | | ✓ | 6 | RMSE-norm | 1.983 | 2.131 | 1.905 | 0.814 | 0.816 | 1.831 | **0.517** | 1.499 | 0.545 |
| | | | | | | spearman | 0.956 | 0.955 | 0.926 | **0.995** | 0.995 | 0.813 | 0.806 | 0.811 | **0.817** |
| | | | | | | bias | -2.001 | -2.175 | -1.921 | -0.811 | -0.814 | -1.847 | **-0.068** | -1.471 | 0.391 |

Table 10: Comparing performance measures of MI Estimators: Correlated Gaussian Base

| Transformation | | | | | N | d | measure | KSG-Based Measures | | | | | MINE-Based Measures | | | |
|---|---|---|---|---|---|---|---|---|---|---|---|---|---|---|---|---|
| rm | cb | sg | ds | dn | | | | ksg | bi-ksg | ksg-loc | ksg-glo | ksg-glo-$L_\infty$ | mine | mine-loc | mine-glo | mine-glo-corr |
| | | | | | 200 | 2 | RMSE-norm | 0.125 | 0.125 | 0.122 | 0.113 | 0.115 | 0.573 | 0.579 | 0.601 | 0.555 |
| | | | | | | | spearman | 0.918 | 0.922 | 0.924 | 0.938 | 0.936 | 0.478 | 0.711 | 0.596 | 0.801 |
| | | | | | | | bias | -0.021 | -0.023 | -0.020 | 0.014 | 0.015 | -0.238 | -0.247 | -0.257 | -0.238 |
| | | | ✓ | | 200 | 2 | RMSE-norm | 0.138 | 0.140 | 0.140 | 0.160 | 0.116 | 0.359 | 0.321 | 0.475 | 0.315 |
| | | | | | | | spearman | 0.923 | 0.923 | 0.914 | 0.898 | 0.878 | 0.886 | 0.929 | 0.931 | 0.940 |
| | | | | | | | bias | -0.049 | -0.051 | -0.048 | -0.042 | -0.005 | -0.114 | -0.089 | -0.192 | -0.108 |
| | ✓ | | ✓ | | 200 | 2 | RMSE-norm | 0.305 | 0.332 | 0.281 | 0.324 | 0.233 | 0.850 | 0.416 | 0.544 | 0.405 |
| | | | | | | | spearman | 0.794 | 0.798 | 0.911 | 0.916 | 0.867 | 0.586 | 0.889 | 0.907 | 0.944 |
| | | | | | | | bias | -0.124 | -0.146 | -0.117 | -0.125 | -0.077 | -0.282 | -0.149 | -0.227 | -0.156 |
| | ✓ | ✓ | ✓ | | 1000 | 2 | RMSE-norm | 0.129 | 0.151 | 0.072 | 0.094 | 0.060 | 0.278 | 0.135 | 0.195 | 0.155 |
| | | | | | | | spearman | 0.961 | 0.958 | 0.941 | 0.960 | 0.952 | 0.951 | 0.942 | 0.969 | 0.967 |
| | | | | | | | bias | -0.046 | -0.064 | -0.019 | -0.023 | 0.000 | -0.106 | -0.029 | -0.057 | -0.038 |
| | | ✓ | ✓ | | 1000 | 2 | RMSE-norm | 0.090 | 0.103 | 0.046 | 0.052 | 0.049 | 0.274 | 0.130 | 0.169 | 0.144 |
| | | | | | | | spearman | 0.947 | 0.950 | 0.954 | 0.943 | 0.961 | 0.957 | 0.941 | 0.973 | 0.969 |
| | | | | | | | bias | -0.031 | -0.041 | -0.004 | -0.003 | 0.017 | -0.104 | -0.025 | -0.042 | -0.032 |
| | ✓ | | ✓ | | 1000 | 2 | RMSE-norm | 0.126 | 0.147 | 0.065 | 0.060 | 0.060 | 0.489 | 0.281 | 0.358 | 0.290 |
| | | | | | | | spearman | 0.940 | 0.935 | 0.939 | 0.949 | 0.950 | 0.879 | 0.950 | 0.946 | 0.950 |
| | | | | | | | bias | -0.042 | -0.060 | -0.020 | 0.000 | 0.000 | -0.213 | -0.105 | -0.145 | -0.112 |
| | ✓ | | | | 200 | 5 | RMSE-norm | 0.962 | 1.088 | 0.481 | 0.470 | 0.469 | 1.014 | 0.949 | 0.982 | 0.934 |
| | | | | | | | spearman | 0.736 | 0.408 | 0.964 | 0.966 | 0.969 | 0.715 | 0.792 | 0.852 | 0.789 |
| | | | | | | | bias | -0.664 | -0.822 | -0.322 | -0.293 | -0.294 | -0.703 | -0.641 | -0.673 | -0.629 |
| ✓ | | | | | 200 | 5 | RMSE-norm | 0.738 | 0.783 | 0.688 | 0.628 | 0.625 | 0.974 | 0.956 | 0.993 | 0.952 |
| | | | | | | | spearman | 0.713 | 0.569 | 0.893 | 0.895 | 0.905 | 0.606 | 0.804 | 0.765 | 0.833 |
| | | | | | | | bias | -0.442 | -0.498 | -0.406 | -0.349 | -0.346 | -0.672 | -0.648 | -0.681 | -0.649 |
| | ✓ | | | | 200 | 5 | RMSE-norm | 0.778 | 0.819 | 0.726 | 0.695 | 0.693 | 1.084 | 0.980 | 0.999 | 0.967 |
| | | | | | | | spearman | 0.933 | 0.931 | 0.932 | 0.937 | 0.939 | 0.605 | 0.773 | 0.805 | 0.711 |
| | | | | | | | bias | -0.539 | -0.589 | -0.498 | -0.462 | -0.460 | -0.784 | -0.670 | -0.688 | -0.660 |
| | | | | ✓ | 1000 | 5 | RMSE-norm | 0.258 | 0.259 | 0.793 | 0.253 | 0.253 | 0.514 | 0.489 | 0.680 | 0.285 |
| | | | | | | | spearman | 0.990 | 0.989 | 0.812 | 0.985 | 0.986 | 0.974 | 0.847 | 0.968 | 0.954 |
| | | | | | | | bias | -0.149 | -0.152 | -0.486 | -0.134 | -0.134 | -0.304 | 0.171 | -0.441 | 0.076 |
| ✓ | ✓ | | | | 1000 | 5 | RMSE-norm | 0.898 | 0.981 | 0.792 | 0.767 | 0.744 | 0.952 | 0.767 | 0.819 | 0.766 |
| | | | | | | | spearman | 0.582 | 0.433 | 0.943 | 0.963 | 0.965 | 0.511 | 0.923 | 0.924 | 0.902 |
| | | | | | | | bias | -0.523 | -0.615 | -0.472 | -0.456 | -0.440 | -0.680 | -0.526 | -0.567 | -0.533 |
| | | | ✓ | | 200 | 10 | RMSE-norm | 1.381 | 1.473 | 1.008 | 0.999 | 1.000 | 1.441 | 1.357 | 1.424 | 1.352 |
| | | | | | | | spearman | 0.131 | 0.170 | 0.956 | 0.962 | 0.963 | 0.512 | 0.790 | 0.806 | 0.856 |
| | | | | | | | bias | -1.351 | -1.522 | -0.985 | -0.957 | -0.960 | -1.412 | -1.288 | -1.388 | -1.290 |
| | ✓ | ✓ | ✓ | ✓ | 200 | 10 | RMSE-norm | 1.424 | 1.514 | 1.384 | 1.122 | 1.121 | 1.446 | 1.063 | 1.411 | 1.096 |
| | | | | | | | spearman | -0.127 | -0.083 | 0.751 | 0.935 | 0.941 | 0.590 | 0.005 | 0.771 | 0.786 |
| | | | | | | | bias | -1.212 | -1.383 | -1.175 | -0.910 | -0.909 | -1.418 | -0.451 | -1.362 | -0.784 |
| | ✓ | | ✓ | | 200 | 10 | RMSE-norm | 1.356 | 1.412 | 1.229 | 1.321 | 1.213 | 1.287 | 1.163 | 1.382 | 1.162 |
| | | | | | | | spearman | 0.831 | 0.823 | 0.872 | 0.885 | 0.900 | 0.652 | 0.933 | 0.942 | 0.916 |
| | | | | | | | bias | -1.164 | -1.263 | -1.043 | -1.111 | -1.011 | -1.182 | -0.990 | -1.325 | -0.976 |
| ✓ | | | ✓ | | 1000 | 10 | RMSE-norm | 1.217 | 1.260 | 1.158 | 1.353 | 1.063 | 0.937 | 0.886 | 1.077 | 0.912 |
| | | | | | | | spearman | 0.815 | 0.641 | 0.947 | 0.947 | 0.986 | 0.956 | 0.954 | 0.982 | 0.969 |
| | | | | | | | bias | -1.033 | -1.120 | -0.976 | -1.145 | -0.890 | -0.869 | -0.844 | -1.030 | -0.861 |
| | | ✓ | ✓ | | 1000 | 10 | RMSE-norm | 1.424 | 1.516 | 1.386 | 0.875 | 0.877 | 1.185 | 0.808 | 1.090 | 0.723 |
| | | | | | | | spearman | -0.065 | -0.450 | 0.845 | 0.992 | 0.992 | 0.748 | 0.875 | 0.951 | 0.898 |
| | | | | | | | bias | -1.212 | -1.386 | -1.177 | -0.721 | -0.723 | -1.145 | 0.444 | -0.981 | 0.560 |
| ✓ | | | | ✓ | 1000 | 2 | RMSE-norm | 0.141 | 0.159 | 0.093 | 0.078 | 0.061 | 0.280 | 0.157 | 0.235 | 0.150 |
| | | | | | | | spearman | 0.968 | 0.943 | 0.967 | 0.964 | 0.967 | 0.793 | 0.953 | 0.960 | 0.950 |
| | | | | | | | bias | -0.040 | -0.050 | -0.031 | -0.014 | 0.000 | -0.105 | -0.017 | -0.085 | -0.027 |

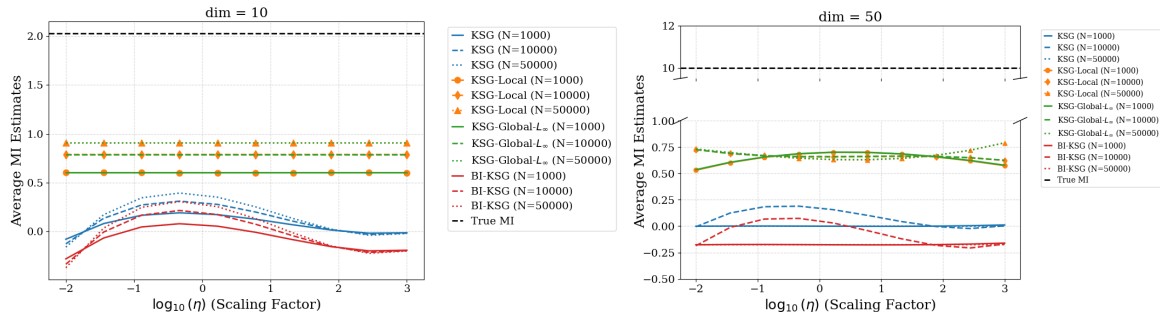

Figure 7: Analysis of the KSG Estimators in response to data scaling for varying number of data samples $N = \{1000, 10000, 50000\}$, and different data dimensionality $d = \{10, 50\}$. Estimates are for $I(\eta X; T)$, where $\eta$ is the scaling factor.

## H    Additional Results

### H.1    Scale Invariance Testing

In the same setting as Section 4.2, we conduct more experiments to see if behaviour of the various estimators in response to data scaling remains unchanged for a different number of sampled datapoints $N = \{1000, 5000\}$ and $d = \{10, 50\}$. The results are shown in Figures 7 and 8.

**KSG:** Overall, for KSG (Figure 7), when we're analyzing the behaviour of the native estimators, we find that they show similar response to scale, i.e., they converge to very low values near zero as the scale $\eta$ goes to either extremity. In contrast, the scale-invariant estimators preserve the response across scales for $d = 10$. For higher dimensionality $d = 50$, we see some interesting behavioral changes for the scale-invariant KSG-Local and KSG-Global variants. We find that although they do not reach negligible values when scale reaches either extreme, the average MI estimates do not stay the same for all $\eta$, and there is some variation. Overall, the KSG-Global and Local variants behave similarly as before, and their average measures are relatively stable when $\eta$ is no not small or too large. Only when $\eta$ is either less than 0.1 or greater than 10, we see a slightly more pronounced change for the average MI estimates.

**MINE:** For the MINE results (Figure 8), we observe some interesting variations in the scale response depending on the dimensionality and the number of datapoints. We find that when we use a greater number of datapoints, then the MINE estimate converges to zero only for smaller scaling factors $\eta$. Our result in Proposition 5 essentially applies for the limiting case when $\eta \to 0^+$, and we can intuitively show that the value of $\eta$ for which we see this limiting behaviour reduces with the greater sample number. The proof of Proposition 5 essentially finds that the weights associated with the scaled RV $\eta T$ is upper bounded in magnitude by a multiplicative factor of $\eta$ and the number of updates and epochs. As a greater number of datapoints also implies a greater number of gradient descent updates, this also implies that we shall have potentially larger weights when the sample number increases. This implies that for larger datapoints $N$ the MI estimate may stay non-zero for a larger range of $\eta$, and drop to near-zero values only near the extreme values of $\eta$, which is what we see in Figure 8. Lastly, we see that the local and global variants of MINE have relatively stable behaviour of their mean values across $\eta$ for all $d, N$ combinations. However, we do see that the variance of these estimators increases with more datapoints.

### H.2    Impact of noise in neural network training

In this experiment, we measure the MI between the inputs ($X$) and the features ($Z$) of a neural network, when trained on the MNIST dataset. An additive noise layer is introduced after the $Z$ layer that adds Gaussian noise of the form $\mathcal{N}(0, \sigma^2)$. Thus, subsequently, $I(X; Z + \mathcal{N}(0, \sigma^2))$ is measured at the last epoch of the training to evaluate how the MI changes when additive Gaussian noise is introduced. In figure 9, the noise level is characterized using the signal-to-noise ratio (SNR) numbers, which quantifies the strength of a signal relative to the background noise. Specifically, an SNR of $a$ dB implies that with unit signal power, the noise variance is $10^{-a/10}$. Consequently, as SNR increases, the noise level decreases, leading to a reduction in

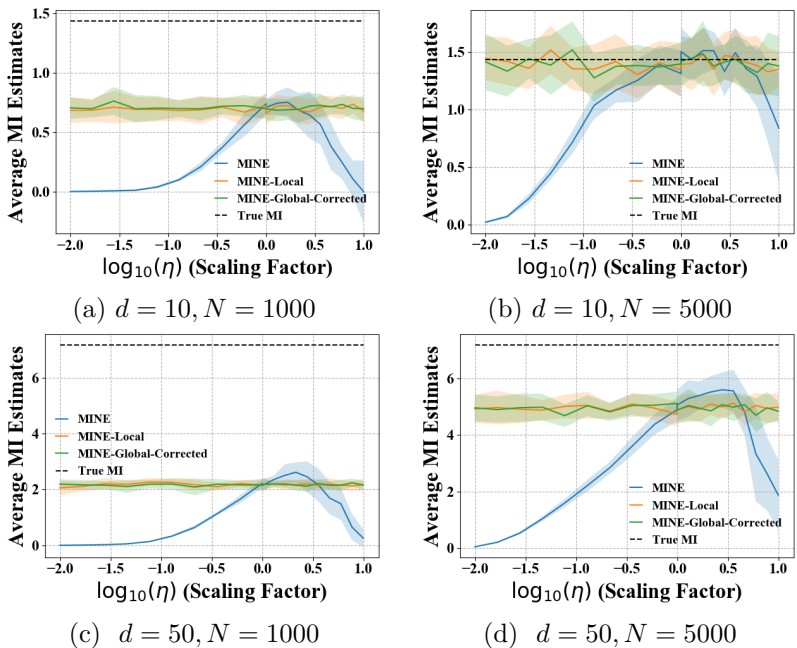

Figure 8: Analysis of the MINE Estimators in response to data scaling for varying number of data samples $N = \{1000, 5000\}$, and different data dimensionality $d = \{10, 50\}$. Estimates are for $I(\eta X; T)$, where $\eta$ is the scaling factor.

interference. The mutual information $I(X; Z + \mathcal{N}(0, \sigma^2))$ is expected to increase as the noise level decreases. Since the SNR is expressed in decibels (dB), the noise reduction is more pronounced within the range of low SNRs (e.g. 0 to 15 dB), decreasing faster compared to the range of high SNRs (15 to 30 dB), and MI within the range of low SNR should also have a higher growing rate compared to high SNRs.

In Figure 9, we present the trend of $I(X; Z + \mathcal{N}(0, \sigma^2))$ during training, comparing the original KSG estimator, its local-normalized and global-normalized variants, and the MINE estimator and its variants. The displayed results represent the averages from 10 trials. For the original KSG estimator, we observe that as the SNR increases, the mutual information does not change as initially anticipated. Instead, the results for the original KSG estimator initially increase and then decline as SNR continues to rise. Notably, both estimators with global normalization exhibit the most consistent trend, reflecting the expected increase in dependence between $X$ and $Z$ as noise is reduced, and having a higher growing rate at low SNR regime.

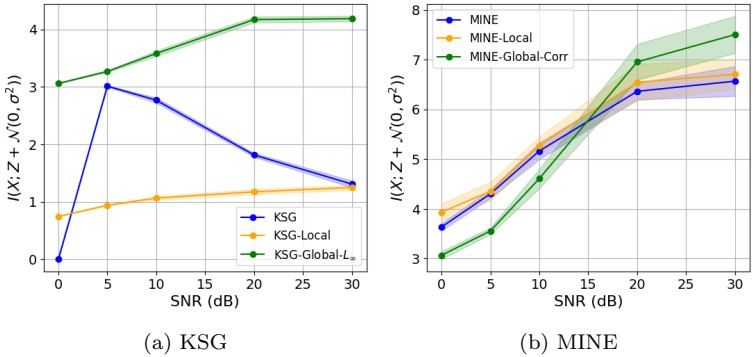

Figure 9: $I(X; Z + \mathcal{N}(0, \sigma^2))$ results with varying SNR. Shaded regions indicate 95% confidence intervals derived from 10 trials.

## I  Information Plane Analysis

In Figure 10, we present the information plane analysis for the IB, MNIST, CIFAR-10 and SVHN datasets, employing both KSG and MINE estimators to examine MI dynamics during neural network training. The displayed curves represent averages from 10 independent trials.

For the IB and MNIST datasets, the original KSG estimator shows that both $I(X;Z)$ and $I(Y;Z)$ generally increase as training progresses. However, our proposed KSG-global-$L_\infty$ estimator reveals a clear information bottleneck pattern. Specifically, it distinctly identifies two phases: a fitting phase characterized by an initial increase in $I(X;Z)$ that eventually stabilizes, and a subsequent compression phase where $I(X;Z)$ decreases while $I(Y;Z)$ continues to increase. For CIFAR-10, the original KSG estimator exhibits an unstable and irregular trend, initially increasing, then decreasing, and subsequently increasing again. We hypothesize that the initial compression phase is later overshadowed by scale-related effects. In contrast, the KSG-global-$L_\infty$ estimator clearly and consistently demonstrates a monotonic compression phase, showing clear differences from MNIST.

Results from experiments with MINE estimators also yield insightful observations. For the IB dataset, the vanilla MINE estimator initially shows an unexpected reduction in both $I(X;Z)$ and $I(Y;Z)$, indicating a brief period where the model seems to discard label information. Subsequently, we see clear fitting and compression phases, but only for the local and global variants. On the MNIST dataset, vanilla MINE captures only the fitting phase, with $I(X;Z)$ continuously increasing and no subsequent compression phase. In contrast, our proposed MINE-global-corrected variant clearly demonstrates both fitting and compression phases, unlike the local variant. For CIFAR-10, vanilla MINE only detects a fitting phase, while local-normalized and global-normalized MINE variants successfully reveal clear compression phases, echoing the results from their KSG counterparts.

In SVHN, there are some interesting observations. First, we see that the vanilla KSG and MINE, and the local normalization variants show inconsistent trends. Notably, the vanilla estimators show very less variation in $I(X;Z)$, thereby not capturing any notable trend. In contrast, the global variants of KSG and MINE show a consistent trend, showing a steady compression in terms of $I(X;Z)$. But interestingly, for $I(Y;Z)$ we observe that both global and local variants capture a phase where $I(Y;Z)$ increases, and then a phase when $I(Y;Z)$ actually decreases, which is reminiscent of the compression phase but for $I(Y;Z)$ instead of $I(X;Z)$.

## J  MI versus Feature Energy in Neural Networks

An important assumption in our work is that individual dimensions of $X$ with lower energy compared to others have less impact on the MI estimate, as they are potentially noisier relative to the target variable. This assumption is central to highlighting the advantages of structure-preserving global normalization over local normalization. In this section, we verify if this is true for neural networks. Here, we analyze $I(z_i;Y)$, where $Z \in \mathbb{R}^d = [z_1, z_2, \ldots, z_d]$ are the convolutional neural network features that are input to the fully connected layers, and $Y$ is the ground truth output label. Our overall hypothesis is, that lower the energy of $z_i$, the less likely it is to share any meaningful information with the ground truth output $Y$, and thus more likely it is to be a noisy dimension w.r.t the output, and can be ignored. To test this hypothesis, we conduct an experiment by training CNNs on MNIST and CIFAR-10. The architecture for MNIST consists of convolutional layers: [64, MaxPool($2\times2$), 128, MaxPool($2\times2$), 256, MaxPool($2\times2$) with padding 1, 512, MaxPool($4\times4$)], followed by a fully connected layer of size 512.

Similarly, for CIFAR-10, the architecture consists of convolutional layers: [64, MaxPool($2\times2$), 128, MaxPool($2\times2$), 256, MaxPool($2\times2$), 512, MaxPool($4\times4$)], followed by a fully connected layer of size 512.

In both cases, we define $Z$ as the input to the fully connected layer, making $Z \in \mathbb{R}^{512}$ a 512-dimensional random variable. To analyze the relationship between feature energy and mutual information, we compute the MI between each individual feature dimension $z_i$ and the ground truth label $Y$. Simultaneously, we measure the average energy of each feature dimension as: $a_i = \sum_{i=1}^n \frac{z_i^2}{n}$, where $n$ is the total number of data samples.

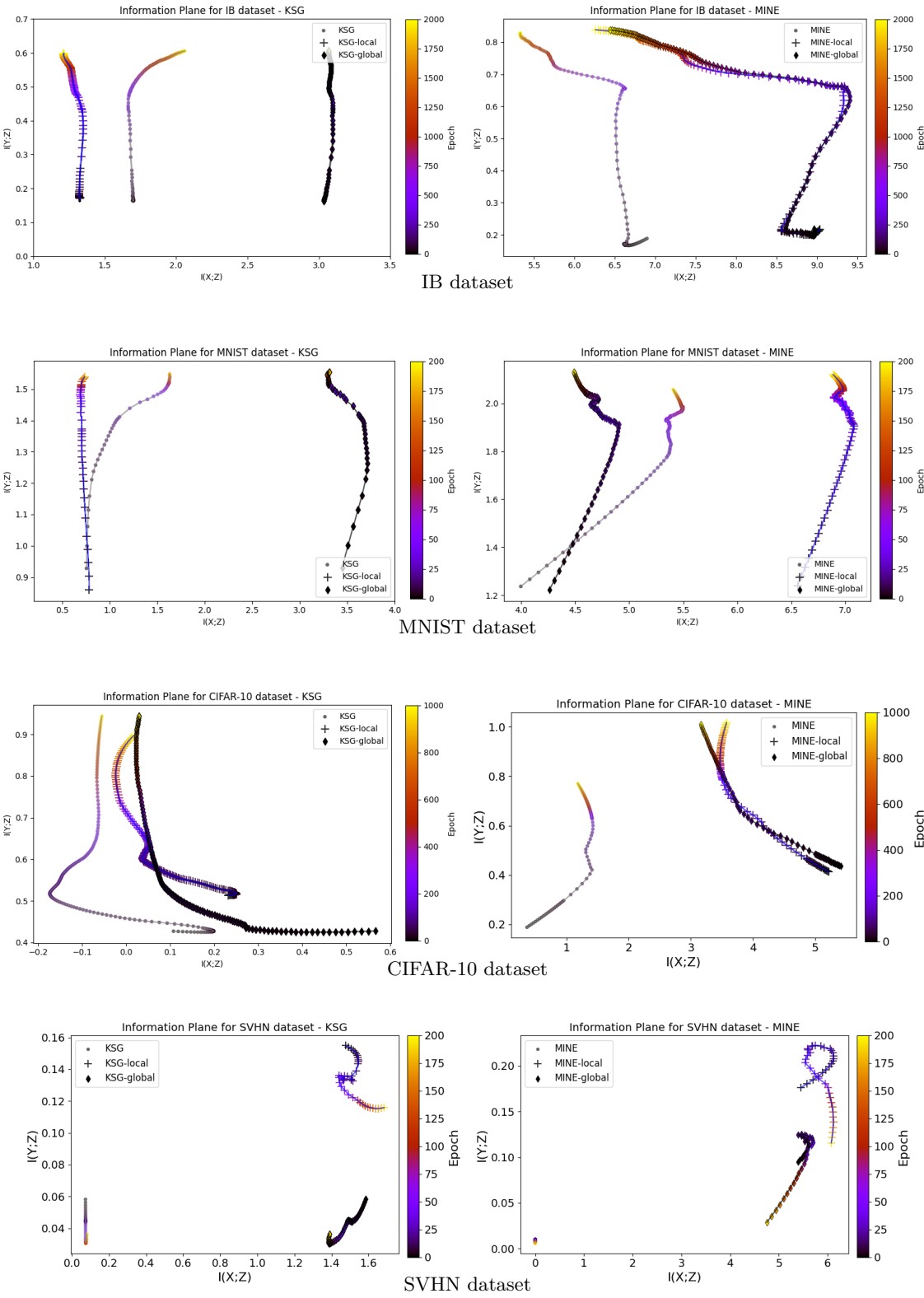

Figure 10: Information plane ($I(X; Z)$ against $I(Y; Z)$) for IB, MNIST, CIFAR-10 and SVHN datasets. $Z$ is the output of $3^{rd}$ layer for IB dataset and MNIST dataset, and $7^{th}$ layer for CIFAR-10 and SVHN datasets. Details in Appendix F.3

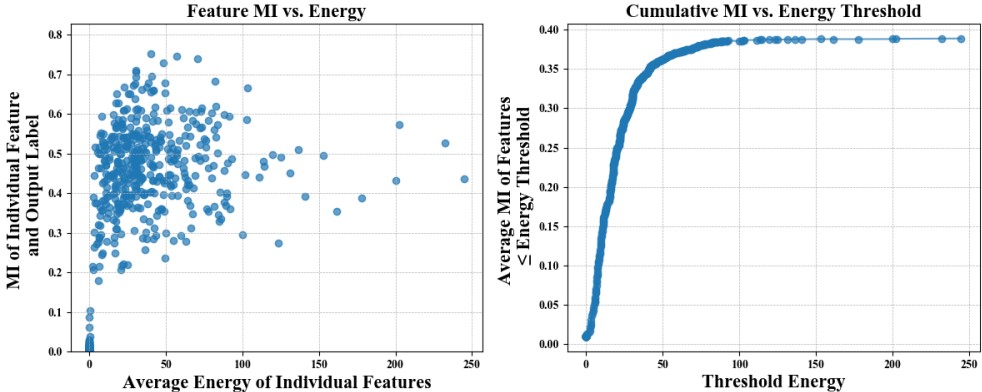

(a) MNIST (left: Scatter plot of $a_i$ vs. $I(z_i, Y)$ and right: threshold-based MI plot).

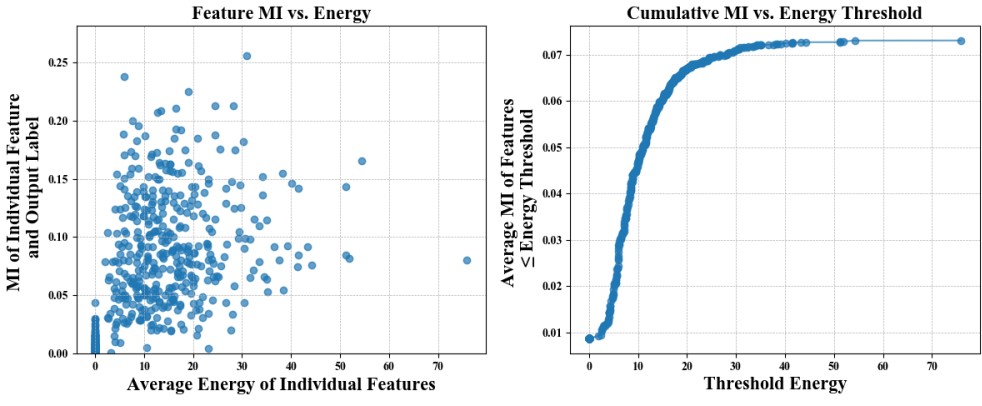

(b) CIFAR-10 (left: Scatter plot of $a_i$ vs. $I(z_i, Y)$ and right: threshold-based MI plot.)

Figure 11: Comparison of individual feature dimension energy and its mutual information with the labels for MNIST and CIFAR-10.

We present two key results: (1) a scatter plot of $a_i$ against the corresponding MI $I(z_i, Y)$, illustrating the relationship between feature energy and information content, and (2) a plot showing the average MI of feature dimensions with energy below a threshold $\tau$, computed as $\mathbb{E}_{a_i \leq \tau}[I(z_i, Y)]$, with $\tau$ varying along the x-axis. The results are summarized in Figure 11.

Our findings are clear: lower the average energy of a feature, less relevant it is to the model's output. This reinforces our hypothesis that in the neural network setting, lower energy features are more likely to behave as noise in the context of MI estimation.

## K Proofs of theoretical results

**Proposition 1.** Let $\widehat{I}_{\text{bin}}^n(X; T)$ denote the mutual information estimated using a fixed number of bins per dimension, with bin edges defined by the minimum and maximum values of the data. Then, for all $\alpha \in \mathbb{R}^+$,

$$\widehat{I}_{\text{bin}}^n(\alpha X; \alpha T) = \widehat{I}_{\text{bin}}^n(X; T) \quad \& \quad \widehat{I}_{\text{bin}}^n(X; \alpha T) = \widehat{I}_{\text{bin}}^n(X; T).$$

*Proof.* For $X \in \mathbb{R}^d$, define $X_{\min}, X_{\max} \in \mathbb{R}^d$ as the componentwise extrema. Under scaling by $\alpha > 0$, we have:

$$\min(\alpha X) = \alpha X_{\min} \quad \& \quad \max(\alpha X) = \alpha X_{\max},$$

so the bin edges scale by $\alpha$, and the number of bins remains fixed. Thus, each bin in $X$ corresponds bijectively to a bin in $\alpha X$ with identical counts. Therefore, the empirical distribution over bins, and hence $\widehat{I}^n_{\text{bin}}$, is invariant under such scaling. □

**Proposition 2.** It holds that $\widehat{I}^n_{KSG}(\alpha X; \alpha T) = \widehat{I}^n_{KSG}(X; T)$, $\forall \alpha \in \mathbb{R}^+$.

*Proof.* Let $\psi$ denote the digamma function(Abramowitz, 1974), and

$$n_{\alpha x, i, \infty} = \sum_{j \neq i} \mathbb{I}\{||\alpha X_i - \alpha X_j||_\infty \leq \rho_{k, i, \infty}\}, \tag{10}$$

where $\rho_{k, i, \infty}$ is the sup-norm distance of $\{\alpha X_i, \alpha T_i\}$ to its k-nearest neighbor in the joint $\{\alpha X, \alpha T\}$ space. Here $||\alpha X_i - \alpha X_j||_\infty$ represents the $X$-dimensions only distance.

Then, the KSG estimate of mutual information (equation 3 from (Kraskov et al., 2004) is

$$\widehat{I}^n_{KSG}(\alpha X; \alpha T) = \psi(k) + \psi(n) - \frac{1}{k} - \frac{1}{n} \sum_{i=1}^n \left( \psi(n_{\alpha x, i, \infty}) + \psi(n_{\alpha t, i, \infty}) \right). \tag{11}$$

Let $\rho'_{k, i, \infty}$ denote the sup-norm distance of $\{X_i, T_i\}$ to its k-nearest neighbor in the joint $\{X, T\}$ space for the unscaled variables. It is trivial to see that $\rho_{k, i, \infty} = \alpha \rho'_{k, i, \infty}$.

We then have,

$$n_{\alpha x, i, \infty} = \sum_{j \neq i} \mathbb{I}\{||\alpha X_i - \alpha X_j||_\infty \leq \alpha \rho'_{k, i, \infty}\} = \sum_{j \neq i} \mathbb{I}\{||X_i - X_j|| \leq \rho'_{k, i, \infty}\} = n_{x, i, \infty}. \tag{12}$$

One can similarly show that $n_{\alpha t, i, \infty} = n_{t, i, \infty}$.

Thus, $\psi(n_{\alpha x, i, \infty}) = \psi(n_{x, i, \infty})$ and $\psi(n_{\alpha t, i, \infty}) = \psi(n_{t, i, \infty})$, and it follows that $\widehat{I}^n_{KSG}(\alpha X; \alpha T) = \widehat{I}^n_{KSG}(X; T)$. □

**Proposition 3.** Let $\{(X_i, T_i)\}_{i=1}^n$ be drawn i.i.d. from a bounded distribution on $\mathbb{R}^{d_X} \times \mathbb{R}^{d_T}$ that is absolutely continuous. Then, it holds almost surely that

$$\lim_{\alpha \to 0^+} \widehat{I}^n_{KSG}(X; \alpha T) = \lim_{\alpha \to \infty} \widehat{I}^n_{KSG}(X; \alpha T) = -\frac{1}{k}.$$

Thus, $\widehat{I}^n_{KSG}(X; \alpha T)$ need not be equal to $\widehat{I}^n_{KSG}(X; T)$.

*Proof.* Following the proof of Proposition 2, let $\rho_{k, i, \infty}$ denote the sup-norm distance from $(X_i, \alpha T_i)$ to its $k$-th nearest neighbor in the joint space $(X, \alpha T)$. Define

$$n_{\alpha t, i, \infty} = \sum_{j \neq i} \mathbb{I}\{||\alpha T_i - \alpha T_j||_\infty \leq \rho_{k, i, \infty}\}, \tag{13}$$

and similarly for $n_{x, i, \infty}$. The KSG estimator's estimate here will be:

$$\widehat{I}^n_{KSG}(X; \alpha T) = \psi(k) + \psi(n) - \frac{1}{k} - \frac{1}{n} \sum_{i=1}^n \left( \psi(n_{x, i, \infty}) + \psi(n_{\alpha t, i, \infty}) \right). \tag{14}$$

First, we observe that

$$\lim_{\alpha \to 0^+} \left\| (X_i, \alpha T_i) - (X_j, \alpha T_j) \right\|_\infty = \lim_{\alpha \to 0^+} \max\{||X_i - X_j||_\infty, \ \alpha ||T_i - T_j||_\infty\} = ||X_i - X_j||_\infty. \tag{15}$$

Furthermore, as $X$ and $T$ are absolutely continuous, it holds almost surely that the k-nearest neighbor distance in $X$ space will be greater than zero and finite.

In the limit $\alpha \to 0^+$, all pairwise distances $||\alpha T_i - \alpha T_j||_\infty \to 0$, and thus $n_{\alpha t,i,\infty} = \sum_{j \neq i} \mathbb{I}\{||\alpha T_i - \alpha T_j||_\infty \leq \rho_{k,i,\infty}\} \to n$.

Next, from equation 15 it follows that $\rho_{k,i,\infty}$ becomes the k-nearest neighbor distance in $X$ space as $\alpha \to 0^+$, and thus we can write

$$\lim_{\alpha \to 0^+} n_{x,i,\infty} = \sum_{j \neq i} \mathbb{I}\{||X_i - X_j||_\infty \leq \rho_{k,i,\infty}\} = k. \tag{16}$$

This enables us to replace the terms in equation 14 yielding

$$\lim_{\alpha \to 0^+} \psi(k) + \psi(n) - \frac{1}{k} - \frac{1}{n} \sum_{i=1}^{n} (\psi(k) + \psi(n)) = -\frac{1}{k}. \tag{17}$$

Lastly, by global scale invariance (Proposition 2), $\widehat{I}_{KSG}^n(X; \alpha T) = \widehat{I}_{KSG}^n(\frac{1}{\alpha} X; T)$, and thus,

$$\lim_{\alpha \to 0} \widehat{I}_{KSG}^n(X; \alpha T) = \lim_{\alpha \to 0} \widehat{I}_{KSG}^n(\frac{1}{\alpha} X; T) = \lim_{\beta \to \infty} \widehat{I}_{KSG}^n(\beta X; T) = -\frac{1}{k}. \tag{18}$$

$\square$

**Proposition 4.** It holds that $\widehat{I}_{MINE-opt}^n(X; \alpha T) = \widehat{I}_{MINE-opt}^n(X; T) \,\forall \alpha \in \mathbb{R}^+$.

*Proof.* Let $f$ be any neural network function used in the MINE objective:

$$\mathbb{E}_{(X,T) \sim P(X,T)}[f(X,T)] - \log \mathbb{E}_{(X,T) \sim P(X) \times P(T)}[e^{f(X,T)}]. \tag{19}$$

Define a corresponding function $f'$ for the scaled variable $\alpha T$ by setting the first-layer weights on $T$ in $f'$ to $W_T' = W_T/\alpha$, and keeping all other parameters identical. Then $f'(X, \alpha T) = f(X, T)$, so both expectations in the MINE objective remain unchanged after the transformation.

As this holds for any $f$, the supremum over all such functions is preserved under scaling of $T$. Hence, $\widehat{I}_{MINE-opt}^n(X; \alpha T) = \widehat{I}_{MINE-opt}^n(X; T)$ for all $\alpha > 0$. $\square$

**Proposition 5.** Consider the MINE optimization problem with input data $S = \{(\alpha X_1, Y_1), \ldots, (\alpha X_n, Y_n)\}$ where $X \in \mathbb{R}^{d_x}$, $Y \in \mathbb{R}^{d_y}$, $(X, Y) \sim P(X, Y)$ are bounded RVs and $\alpha \in \mathbb{R}^+$ is a scaling factor. We consider a neural network of depth $L+1$ having $h_1, h_2, \ldots, h_L$ ReLU-activated hidden neurons in the respective layers. The network is trained via gradient descent on the MINE loss function in Belghazi et al. (2018) for a fixed number of iterations $n_T$, with a learning rate schedule $0 \leq \eta(t) < \infty$ for all $t \leq n_T$. Let the weights between the $i^{th}$ node of the $l+1^{th}$ hidden layer and the $j^{th}$ node of the $l^{th}$ hidden layer after $t$ iterations be denoted by $w_{ji}^l(t)$. Assume that the initialized weights are bounded, i.e., $\forall (l, i, j)$, $|w_{ji}^l(0)| \leq \epsilon$ for some $\epsilon > 0$. Lastly, let $w_{ji}^0[X]$ denote the first layer weights that are attached to $X$. We then have, $\forall (i, j)$,

$$\lim_{\alpha \to 0^+} |w_{ji}^0[X](n_T)| \leq \epsilon. \tag{20}$$

*Proof.* Let $Z_j^l(\alpha x, y; t)$ denote the output of the $j^{th}$ node at layer $l$ in response to $(\alpha x, y)$ as the input at iteration $t$, where $(x, y)$ is a sampled instance of $P(X, Y)$. Note that when $l = 0$, $Z_j^l$ will be the inputs to the network $(\alpha x, y)$ itself. As we are given the dataset $S$, the weight update rule for $w_{ji}^l$, at iteration $t$ is then $w_{ji}^l(t+1) = w_{ji}^l(t) + \Delta w_{ji}^l(t)$, where

$$\Delta w_{ji}^l(t) = -\eta(t) \frac{1}{n} \sum_{k=1}^{n} Z_j^l(\alpha X_k, Y_k; t) \delta_i^{l+1}(\alpha X_k, Y_k; t), \tag{21}$$

Here, $\delta_i^{l+1}(\alpha X_k, Y_k; t)$ denotes the backpropagation error signal at the $i^{\text{th}}$ neuron of the $(l+1)^{\text{th}}$ layer at iteration $t$, computed in response to the input pair $(\alpha X_k, Y_k)$. For notational simplicity, we omit the input instance arguments in the following expressions, and write the error signal as $\delta_i^{l+1}(t)$. Let $a(\cdot)$ be the ReLU activation with its derivative $a'(\cdot) \in \{0,1\}$. Then we have,

$$\delta_j^{l-1}(t) = a'(z_j^{l-1}(t)) \left( \sum_{i=1}^{h_l} \delta_i^l(t) w_{ji}^{l-1}(t) \right), \tag{22}$$

where $z_j^{l-1}(t)$ denotes the pre-activation output of the $j^{th}$ node of the $(l-1)^{th}$ layer itself. Note that $Z_j^l(t) = a(z_j^l(t))$ for $l \geq 1$. As $|a'| \leq 1$, this yields:

$$|\delta_j^{l-1}(t)| \leq \sum_{i=1}^{h_l} |\delta_i^l(t) w_{ji}^{l-1}(t)|. \tag{23}$$

Let us assume the whole network function can be denoted as $f_W(X, Y) : \mathbb{R}^{d_x + d_y} \to \mathbb{R}$. Note that the network outputs a single real number, and the last layer does not have any activation function (which is ReLU for the other layers). In the context of MINE's optimization problem, the network minimizes the following loss function:

$$\widehat{I}_{\text{MINE}}(X; Y) = -\frac{1}{n} \sum_{i=1}^n f_W(\alpha X_i, Y_i) + \log \left( \frac{1}{n} \sum_{i=1}^n e^{f_W(\alpha X_i, \tilde{Y}_i)} \right). \tag{24}$$

The error signal at the last layer, $\delta^{L+1}(t)$ is the derivative of the loss w.r.t the network output. However, as the MINE optimization effectively has two distributions $P(X, Y)$ and $P(X)P(Y)$ of input, we consider the error signal of these distributions separately. The loss function for the input $X_j, Y_j \sim P(X, Y)$ is $-\frac{1}{n} f_W(\alpha X_j, Y_j)$, which yields an error signal $\delta^{L+1}(t) = -1/n$. Whereas, the loss function for the input $X_j, \tilde{Y}_j \sim P(X)P(Y)$ is $\log \left( \frac{1}{n} \sum_{i=1}^n e^{f_W(\alpha X_i, \tilde{Y}_i)} \right)$, which yields an error signal

$$\delta^{L+1}(t) = \frac{d \left( \log \left( \frac{1}{n} \sum_{i=1}^n e^{f_W(\alpha X_i, \tilde{Y}_i)} \right) \right)}{d f_W(\alpha X_j, \tilde{Y}_j)} = \frac{e^{f_W(\alpha X_j, \tilde{Y}_j)}}{\sum_{i=1}^n e^{f_W(\alpha X_i, \tilde{Y}_i)}} \leq 1. \tag{25}$$

Thus, across both cases, we have that the error signal $|\delta^{L+1}(t)| \leq 1$.

Next, we consider the case when $|\alpha| \leq A$. where $A \in \mathbb{R}^+$ is a finite real. Within this setting, using the principle of induction, we now will show that the error signal $|\delta_j^l(t)| \leq D_t \ \forall (l, j)$ and the weights $|w_{ji}^l(t)| \leq B_t$ $\forall (l, i, j)$ for some finite reals $D_t, B_t$ which are independent of $\alpha$.

First, let us assume weights at iteration $t$ satisfy $|w_{ji}^l(t)| \leq B_t \ \forall (l, i, j)$ for some $B_t \in \mathbb{R}^+$, which is independent of $\alpha$. Then, it first follows from equation 23 that

$$|\delta_j^{l-1}(t)| \leq \sum_{i=1}^{h_l} B_t |\delta_i^l(t)|, \tag{26}$$

and as $|\delta^{L+1}(t)| \leq 1$, this ultimately yields

$$|\delta_j^l(t)| \leq (B_t)^{L+1-l} \prod_{v=l+1}^L h_v. \tag{27}$$

If we set $D_t = \max_l\{(B_t)^{L+1-l} \prod_{v=l+1}^L h_v\}$, we have that $|\delta_j^l(t)| \leq D_t \ \forall (l, j)$. Note that here $D_t$ only depends on $B_t$ and the network architecture. Following the weight update rule at iteration $t$ from equation 21, we

have:

$$|w_{ji}^l(t+1)| = \left| w_{ji}^l(t) - \frac{\eta(t)}{n} \sum_{k=1}^{n} Z_j^l(\alpha X_k, Y_k; t) \delta_i^{l+1}(t) \right|,$$

$$\leq |w_{ji}^l(t)| + \frac{\eta(t)}{n} \sum_{k=1}^{n} \left| Z_j^l(\alpha X_k, Y_k; t) \right| \left| \delta_i^{l+1}(t) \right|,$$

$$\leq B_t + \eta(t) D_t \left( \frac{1}{n} \sum_{k=1}^{n} \left| Z_j^l(\alpha X_k, Y_k; t) \right| \right). \tag{28}$$

Next, as $X$ and $Y$ are bounded, we can assume that every dimension $|x_i| \leq K$ and $|y_i| \leq K$ for some $K \in \mathbb{R}^+$. Now, as $|\alpha| \leq A$, the inputs $\alpha X$ and $Y$ are bounded. Furthermore, as the weights $|w_{ji}^l(t)| \leq B_t$, the outputs of the layers $|Z_j^l(\alpha X_k, Y_k; t)|$ will be bounded by a finite scalar which can be written as a fixed function of $A$, $B_t$, $K$ and the hidden layer counts $h_1, \ldots, h_L$. Thus,

$$|w_{ji}^l(t+1)| \leq B_t + \eta(t) D_t g(A, B_t, K, h_1, \ldots, h_L). \tag{29}$$

We can set $B_{t+1} = B_t + \eta(t) D_t g(A, B_t, K, h_1, \ldots, h_L)$, and the above then implies that $|w_{ji}^l(t+1)| \leq B_{t+1}$ $\forall(l, i, j)$, where $B_{t+1}$ is also independent of $\alpha$.

Lastly, as the weights at iteration 0 are finite ($|w_{ji}^l(0)| \leq \epsilon$), via the principle of induction we have that for any finite $t$, $|w_{ji}^l(t)| \leq B_t$ for some finite $B_t$, when $|\alpha| \leq A$. Let us consider $B = \max_{t \in \{0,1,\ldots,n_T\}} B_t$. Then for the entire duration of training and for any $|\alpha| \leq A$, we have $|w_{ji}^l(t)| \leq B$. Note that $B$ is independent of $\alpha$.

Thus, plugging this into equation 23, we obtain, $\forall(l, j, t)$,

$$|\delta_j^{l-1}(t)| \leq \sum_{i=1}^{h_l} B |\delta_i^l(t)|. \tag{30}$$

Subsequently, following the fact that $\left| \delta^{L+1}(t) \right| \leq 1$, we get

$$\left| \delta_i^1(t) \right| \leq B^L \prod_{v=2}^{L} h_v. \tag{31}$$

Let $C = KB^L \prod_{v=2}^{L} h_v$. With this, following equation 21, we have that $|\Delta w_{ji}^0[X](t)| \leq \eta(t)\alpha C$. This yields

$$|w_{ji}^0[X](n_T)| \leq \left| w_{ji}^0[X](0) \right| + \alpha \left( \sum_{t=1}^{n_T} |\eta(t)C| \right), \tag{32}$$

$$\leq \epsilon + \alpha C \left( \sum_{t=1}^{n_T} \eta(t) \right). \tag{33}$$

Thus, when $\alpha \to 0^+$, we have that $\epsilon + \alpha C \left( \sum_{t=1}^{n_T} \eta(t) \right) \to \epsilon$, yielding $\lim_{\alpha \to 0^+} \left| w_{ji}^0[X](n_T) \right| \leq \epsilon$. □

**Proposition 6.** We consider the same setting as Proposition 5 for the MINE estimation problem. There, it holds that $\lim_{\alpha \to 0^+} \widehat{I}_{MINE-sgd}^n(X; \alpha T) = 0$ . Thus, $\widehat{I}_{MINE-sgd}^n(X; \alpha T)$ need not be equal to $\widehat{I}_{MINE-sgd}^n(X; T)$.

*Proof.* Let $f^*$ denote the neural network function learned via stochastic gradient descent (SGD) that maximizes the following MINE objective for $\widehat{I}_{MINE-sgd}^n(X; \alpha T)$:

$$\mathbb{E}_{X,\alpha T \sim P(X,\alpha T)}\left[f^*(X, \alpha T)\right] - \log \ \mathbb{E}_{X,\alpha T \sim P(X) \times P(\alpha T)}\left[e^{f^*(X,\alpha T)}\right].$$

Let $W_T$ be the first-layer weights in $f^*$ connected to $\alpha T$. From Proposition 5, as $\alpha \to 0^+$, we have $|W_T|_{ij} \leq \epsilon$, thus yielding finite weights. Hence, the contribution of $\alpha T$ to the output of a neuron of the first hidden layer of $f^*$ becomes $\alpha T \epsilon d_T \to 0$ as $\alpha \to 0^+$, where $d_T$ is the dimensionality of $T$. Therefore, $f^*(X, \alpha T) = f^*(X)$ becomes independent of $T$ in the limit $\alpha \to 0$.

In this case, both expectations in the above objective reduce to empirical averages of the same function $f^*(X)$, and Jensen's inequality implies

$$\frac{1}{n}\sum f^*(X_i) - \log\left(\frac{1}{n}\sum e^{f^*(X_i)}\right) \leq 0,$$

with equality if and only if $f^*(X)$ is constant. Thus, the maximized objective converges to zero.

Hence, $\lim_{\alpha \to 0^+} \widehat{I}^n_{MINE\text{-}sgd}(X; \alpha T) = 0$, showing that $\widehat{I}^n_{MINE\text{-}sgd}(X; \alpha T)$ need not equal $\widehat{I}^n_{MINE\text{-}sgd}(X; T)$. □

**Proposition 7.** We consider the same setting as Proposition 5 for the MINE estimation problem. There, it holds that $\lim_{\alpha \to 0^+} \widehat{I}^n_{MINE-sgd}(X; \alpha T) = 0$ . Thus, $\widehat{I}^n_{MINE-sgd}(X; \alpha T)$ need not be equal to $\widehat{I}^n_{MINE-sgd}(X; T)$.

*Proof.* Let $f^*$ denote the neural network function learned via stochastic gradient descent (SGD) that maximizes the following MINE objective for $\widehat{I}^n_{MINE\text{-}sgd}(X; \alpha T)$:

$$\mathbb{E}_{X,\alpha T \sim P(X,\alpha T)}\left[f^*(X, \alpha T)\right] - \log \ \mathbb{E}_{X,\alpha T \sim P(X) \times P(\alpha T)}\left[e^{f^*(X,\alpha T)}\right].$$

Let $W_T$ be the first-layer weights in $f^*$ connected to $\alpha T$. From Proposition 5, as $\alpha \to 0^+$, we have $|W_T|_{ij} \leq \epsilon$, thus yielding finite weights. Hence, the contribution of $\alpha T$ to the output of a neuron of the first hidden layer of $f^*$ becomes $\alpha T \epsilon d_T \to 0$ as $\alpha \to 0^+$, where $d_T$ is the dimensionality of $T$. Therefore, $f^*(X, \alpha T) = f^*(X)$ becomes independent of $T$ in the limit $\alpha \to 0$.

In this case, both expectations in the above objective reduce to empirical averages of the same function $f^*(X)$, and Jensen's inequality implies

$$\frac{1}{n}\sum f^*(X_i) - \log\left(\frac{1}{n}\sum e^{f^*(X_i)}\right) \leq 0,$$

with equality if and only if $f^*(X)$ is constant. Thus, the maximized objective converges to zero.

Hence, $\lim_{\alpha \to 0^+} \widehat{I}^n_{MINE\text{-}sgd}(X; \alpha T) = 0$, showing that $\widehat{I}^n_{MINE\text{-}sgd}(X; \alpha T)$ need not equal $\widehat{I}^n_{MINE\text{-}sgd}(X; T)$. □

