# OpenReview forum: "Towards Robust Scale-Invariant Mutual Information Estimators"
_TMLR — Accepted by TMLR_

### Review · Reviewer_tz8y · 2025-04-10

**Summary Of Contributions:**

This paper proposes several extensions for the KSG and MINE mutual information estimators that are less sensitive to the scale of variables. The authors first theoretically show that most estimators are not invariant against scale. Next, they conduct multiple empirical studies to discuss the pros and cons of global and local normalization methods. Based on the global normalization method, the authors further adopt 1) the $L_\infty$ norm as a more robust measure of distance for the KSG estimator, and 2) corrected weights for MINE so that it's invariant to dimensionality. The proposed methods are compared on synthetic and real-world estimation tasks, showing superior accuracy over the baseline methods.

**Audience:**

Yes

**Broader Impact Concerns:**

No concerns.

**Claims And Evidence:**

Yes

**Requested Changes:**

* About the definition of normalization methods, Definitions 1-3: I think it's a bit weird that the numerator part of these definitions is not minused by $\mathbb{E}_S[x_i]$, which ensures that the results center at $0$. This differs from the conventional definitions. I understand that KSG is invariant to such kinds of bias, but it's not immediately clear how this would affect the results of MINE. Also, the authors are using the expectation form for the variance term, not the empirical average of samples. Since the variance term is not directly acquirable in practice when data distributions are unknown, this choice may need further justification. Please clarify the differences between your definitions and the conventional one $\frac{x_i - \frac{1}{n} \sum_j x_j}{\sqrt{\frac{1}{n} \sum_j (x_j - \frac{1}{n} \sum_k x_k)^2}}$, and discuss how this would affect the empirical results.
* In Section 5.1.1, the authors discuss the advantage of global normalization over local normalization. While I agree with your point that global norm is more robust to noise components, it is also true that global norm will result in bias towards large-scale components when the informative components are at different scales. In this synthetic study, the authors use isotropic Gaussian to generate the samples, which does not reveal this flaw. Please incorporate this point into your discussion. Also, it may be beneficial to show empirically that whether the informative components of neural features are likely to be of the same scale, such that the results of global norm are unlikely to be biased.
* I would suggest the authors, that besides plotting the estimated $I(X;Z)$, apply their methods in some learning settings, e.g., minimizing $I(X;Z)$ as a regularization, and see how the performance is compared to baseline methods like variational information bottleneck. This would be strong support for the efficacy of the proposed methods.

Minor points:
* Section 5.2.1, the Performance Measures paragraph: I don't quite understand why the ground truth values $\mu_i$ are different across the $k$ trials. If we look at the ds task, since the distribution of $X$ is not changed, the result should be exactly the same. Is this because you are selecting different $\rho$ or $\sigma$ in each trial? If yes, this should be clarified in the paper.
* The proof of Proposition 3: I don't see where the condition $n \to \infty$ is used in the proof. Maybe this conclusion holds for any $n$? Also, the $+$ on the left of $1/n$ should be $-$ in the line below eq. (11).

**Strengths And Weaknesses:**

Strength:
* The paper is well-written and easy to follow. I've checked the main empirical/theoretical claims and the proofs. They all look good to me.
* The algorithm design is well-motivated by the empirical and theoretical findings. Extensive synthetic and real-world experiments clearly demonstrate the superiority of the proposed methods.

Weakness:
* Some places need further clarifications (please refer to the section below).

---

> ### Author Response · Authors · 2025-05-11
> **Responses**
>
> We thank the reviewer for their useful and insightful comments, and we share our responses and updates below.
>
> **Requested Changes** :
>
> 1. **Mean-centering during normalization:** Thank you for pointing this out, we actually mean-center all of our normalization approaches, which are summarized in the `normalize_functions.py` file in the code. We have now updated the definitions accordingly. Also, as suggested, we have clarified the expectations to be the empirical average across samples, which is how we compute them.
> 2. **Correlation between feature scale and relevance:** Yes, we based this assumption on the observation that most studies find that pruning the low energy features of a neural network can still preserve performance (e.g. Dropnet [1]). However, to more concretely test this, we performed an experiment where we measure the MI of individual feature nodes at the output of the feature layers of a trained CNN with the ground truth labels, and then compared this with the overall activation energy of the feature across samples. Interestingly, we see a very convincing trend that shows that low energy features usually also have lower MI with the labels and vice-versa. The experiment has been discussed in Appendix J and the results are shown in Figure 11 of the updated paper. The code has been updated.   Experiments were done on MNIST and CIFAR-10 with 6-layered CNNs.
> 3. **Using I(X;Z) as a regularizer:** Given the primary focus of our paper, which is establishing scale-invariant mutual information estimators and analyzing their theoretical and empirical properties, we believe that incorporating a full-fledged learning-based evaluation and comparison with benchmarks would extend beyond the intended scope of this study. However, we definitely think this is the next step of our work, to validate our normalization approaches being used in conjunction with methods like Infomax [2] to improve generalization further.
>
> **Minor points**:
> 1. Clarifying variation across trials: Yes, the ground truth MI can be different across trials because we’re generating $\rho$ and $\sigma$ randomly each time. We have clarified this in the updated version (Appendix E).
> 2. On $n$ in Proposition 3: Thank you for this observation! Our initial proof was actually slightly different, and there we had to impose the constraint that $n\rightarrow \infty$. You are correct, our current result actually holds for any $n\geq k$ (where $k$ is the $k$-nearest neighbor parameter in KSG estimation), primarily because of the other limit imposed on $\alpha$. This has been now updated.
>
> [1] Tan, C. M. J., & Motani, M. (2020, November). Dropnet: Reducing neural network complexity via iterative pruning. In International conference on machine learning (pp. 9356-9366). PMLR.
>
> [2] R Devon Hjelm, Alex Fedorov, Samuel Lavoie-Marchildon, Karan Grewal, Phil Bachman, Adam Trischler, & Yoshua Bengio (2019). Learning deep representations by mutual information estimation and maximization. In International Conference on Learning Representations.

---

### Review · Reviewer_4xJv · 2025-04-13

**Summary Of Contributions:**

The paper addresses an mportant challenge in mutual information (MI) estimation: sensitivity to scale. The authors show that popular estimators like KSG and MINE are not scale-invariant and propose new, easy-to-implement, estimator-specific normalization strategies that encourage scale-invariance. In addition to improving robustness to diffeomorphic transforms, the scale-invariant estimators are also shown to impact learning dynamics in neural classifiers. Overall, the paper makes a valuable contribution that underscores the importance of scale-awareness in MI estimation, but the work can benefit from a more streamlined experimental section and better embedding in the related work.

**Audience:**

Yes

**Claims And Evidence:**

Yes

**Requested Changes:**

**Major**

- Please, improve the presentation of results and clarify relations to existing work in the main text.
- Please, argue why Gaussians are sufficient for the numerical experiments in sections 5.1 and 5.2 and/or discuss generalizability to non Gaussian distributions (e.g., coupled moments, multimodality).
-  Please, address minor points and questions below, and discuss the limitations of the estimators / experimental setup beyond the linear increase in computation time (e.g., none of the estimators seem generally robust to arbitrary diffeomorphisms, low SNRs remain problematic and can still cause biases).

**Minor**

- The paper would benefit from careful proofreading, as it contains typographical errors (e.g., p.1: $f(Y)$ should be $f(T)$), and some idiosyncrasies (e.g., p.1: “hardness” → “difficulty”) and repeated words.

- In the Motivation section, it doesn’t seem accurate to describe $\alpha$ as “arbitrary,” since the equalities clearly don’t hold when $\alpha = 0$.

- On p.4, equations 7 and 8 are discussed in the context of MINE but are only presented in the appendix.

- “RV” is never defined. Random vector / random variable?

- The notation on p.5 for correlated Gaussians is confusing. The problem is formulated as if $X$ and $T$ are independent, and the setting assumes uncorrelated Gaussians, given that all off-diagonal elements of the covariance matrix are zero.

- The figures are low-resolution and do not make optimal use of space (e.g., large margins, small legends). The captions are also uninformative. Avoid referring to the appendix in figure captions and instead provide a brief, self-contained summary of what each figure demonstrates.

- As the experimental results are dense and the figures are scattered throughout the text, it may be helpful to number the experiments (e.g., Experiment 1: Global vs. Local Normalization) and prepend the experiment label to each figure.

**Questions**

- From the results in Table 3, it appears that all MINE estimators perform poorly on a subset of transformations (last four rows — this effect is also present in Table 2). Do the authors have an explanation for this outcome?

- Are there any implications for incorporating an inductive bias of MI invariance directly into neural network architectures?

**Strengths And Weaknesses:**

**Strengths**
- The paper clearly demonstrates the need to investigate asymptotic and pre-asymptotic scale-invariance properties in MI estimators. My initial instinct was that the problem could be easily addressed via "local" normalization, but the paper convincingly outlines the limitations of this naive approach early on and provides informative experiments on the consequences of different preprocessing choices for the empirical behavior of normalized MI estimators.

- It presents new theoretical results and analyses of popular MI estimators that appear to be missing from the existing literature.

- The empirical evaluation is comprehensive and explores multiple dimensions. The results suggest advantages of the proposed scale-invariant estimators over existing baselines.

- The estimators are straightforward to implement within existing frameworks and have practical implications.

**Weaknesses**
- I found it difficult to delineate the contributions of the paper from established results in Section 3. Are all of these propositions new (as I assume)?

- It seems like an odd choice to place the related work section in the appendix. I believe it’s important to preserve the continuity of research on MI estimators by clearly situating the derivations within the context of recent studies and proposed benchmarks, such as [1]. I am concerned that recent developments are not highlighted with sufficient clarity. For example, p.5 refers to a “recent” variant of KSG, but the cited work is from 2017, which is not recent by deep learning standards. To gain space, other parts of the paper can be streamlined, and figure placement can be highly optimized.

- The experiments involve only Gaussian distributions, which is somewhat misaligned with the generality of the theoretical results and undermines the generalizability of the observed empirical behavior of MI estimators. Moreover, Gaussians have the convenient property that multiplication by a constant isolates changes in variance, whereas other distributions have coupled moments (e.g., Gamma). What about heavy-tailed distributions such as Student-t, where variance depends on both degrees of freedom and scale? Perhaps the authors could briefly discuss how the improved MI estimators could be applied in other practical domains, such as (amortized) optimal design (OD, [2]), where robust MI estimation is crucial.

- The paragraphs and results in the experimental section need to be organized more clearly. There are many single-sentence paragraphs with little connective structure between them.

[1] Czyż, P., Grabowski, F., Vogt, J., Beerenwinkel, N., & Marx, A. (2023). Beyond normal: On the evaluation of mutual information estimators. Advances in Neural Information Processing Systems, 36, 16957-16990.

[2] Rainforth, T., Foster, A., Ivanova, D. R., & Bickford Smith, F. (2024). Modern Bayesian experimental design. Statistical Science, 39(1), 100-114.

---

> ### Author Response · Authors · 2025-05-11
> **Responses**
>
> We thank the reviewer for their detailed, useful and insightful comments, and we share our responses and updates below.
>
> **Weaknesses:**
>
> 1. **Section 3 Results clarification:** Yes, all the results in section 3 are our results and contributions of our work. This has now been clarified in the text.
> 2. **Related works location:** We have currently summarized the discussions on related work in the main text (Section 1.1), which was earlier only present in Appendix B. Also, we have updated the language to more accurately reflect the recency of some of the compared approaches.
> 3. **Studying long-tailed distribution bases:** Yes, our experiments are with Gaussian distribution bases, however, they do go through multiple transformations, some of which can eventually yield more richer variations in the final distribution, such as multimodal behaviour (e.g. sigmoid) and long tailed behaviour as well (e.g. cube). Following your suggestion, we have now also added experiments with Student’s t-distributions, following the same setup as in [1]. We include the same set of transformational configurations as in Tables 2 and 3. The results are given in Table 8 (Page 26) of the updated paper. We summarize our observations as follows. First, we find that for KSG estimators, our proposed variants still outperform other baselines in most configurations, similar to the Gaussian case. However, interestingly we find that MINE estimators struggle with the Student's t-distribution base, and in most cases we see divergent results from the native MINE estimator, yielding very high error. The local and global MINE variants perform better than the vanilla baseline for the initial set of lower dimensional configurations ($d=2,4$), but as soon as dimensionality increases further, they also perform poorly. Nonetheless, we still see that the globally normalized variants of MINE overall perform slightly better than the locally normalized.
> 4. **Experiment presentation**: Yes, following your suggestions we have improved the presentation and writing for the experiments. Details are given below.
>
> **Requested Changes:**
>
> 1. **Presentation of Results:** We have made the following changes to the presentation for greater clarity and better organization.
>       - Experiments have been numbered, and so  have their corresponding figures and tables
>       - A summary of the experiments in each section and their objectives has been included.
>       - Table and Figure placement has been improved
>       - Details for the sampling process for each trial has been separately included in the Appendix for more clarity. (Appendix E)
> 2. **Incorporating heavy tailed distribution bases:**  As per your suggestion, we have conducted experiments with the Student’s t-distribution bases, and added the results and discussions to the paper (Appendix G). Code has been updated.
> 3. **Minor Points and Limitations:** We have gone through all suggested minor improvements and corrections, and have updated the paper accordingly. The conclusion has been shortened and a paragraph on limitations has been added that discusses other limitations of the analyzed estimators and normalization strategies.
>
>
> **Questions:**
>
> 1. **Estimator Performance on the last four configurations:** This is mainly due to higher dimensionality of the data (base distribution $d$). We find that as data dimensionality increases performance suffers significantly. It is also worth mentioning that the final data dimension of the random variables after the transformations can be much greater, as two of our transformations (duplicate-self and duplicate-noise) multiplicatively increase dimensionality by 20-fold in our synthetic experiments. This is also observed with our latest experiments with the Student’s t-distribution base.
> 2. **Can MI invariance be incorporated within NN training as an inductive bias:** This is an interesting question, and yes, we believe that there are some indirect implications of our work which can be potentially used as an inductive bias during NN training. The clearest example of this is with batch normalization. Batch normalization has been developed as a tool for accelerating NN training, and in some cases yields better generalization performance as well. Batch normalization is roughly an analogue to the local normalization approach for pre-processing, and thus it is more likely to break the intrinsic structure of information, as a tradeoff for quicker learning. However, our findings suggest that a global form of the batch normalization, rather than a local form, may be beneficial as well from a structure preserving and MI preserving perspective. We will look into this as potential future work.
>
>
>
> [1] Czyż, P., Grabowski, F., Vogt, J., Beerenwinkel, N., & Marx, A. (2023). Beyond normal: On the evaluation of mutual information estimators. Advances in Neural Information Processing Systems, 36, 16957-16990.

---

> > ### Comment · Reviewer_4xJv · 2025-05-31
> >
> > I thank the authors for their clarifications, edits, and additional experiments. My points have been sufficiently addressed.

---

### Review · Reviewer_1DxP · 2025-04-27

**Summary Of Contributions:**

This work studies the overlooked issue of scale sensitivity in mutual information (MI) estimators, particularly focusing on the popular KSG and MINE methods. The authors show both theoretically and empirically that existing MI estimators are not scale-invariant, meaning that simple rescaling of variables can significantly distort MI estimates. To solve this, they propose new global normalization strategies.

**Audience:**

Yes

**Claims And Evidence:**

Yes

**Requested Changes:**

1. Please discuss other MI estimators.


2. Some larger real-world dataset experiments should be included.

**Strengths And Weaknesses:**

Strengths

1. The paper addresses a subtle yet overlooked issue — scale sensitivity — that had not been deeply studied before in MI estimation. It provides both formal theoretical arguments and empirical evidence to support the claims.

2. The proposed global normalization strategies are practical and improve estimator robustness. Benchmarking across multiple distributions and settings shows the generality and effectiveness of the proposed methods.


Weaknesses


1. Although KSG and MINE are popular, the study focuses mainly on these two, leaving open questions about the generalization to other MI estimators. For example, the InfoNCE estimator is also widely used.



2. While experiments are extensive, it’s unclear if the methods perform equally well in larger datasets like ImageNet.

---

> ### Author Response · Authors · 2025-05-11
> **Responses**
>
> We thank the reviewer for their interesting and useful suggestions. Here are our responses and updates:
>
> **Weaknesses and Requested Changes:**
> 1. **Discussions on other MI Estimators:** Yes, we have currently added some discussions on the one-sided invariance of the InfoNCE estimator (Appendix B.4). Overall, we find that InfoNCE (as proposed by the original authors) is not one sided scale invariant, but can be rendered scale invariant in a similar manner to KSG and MINE by either local or global normalizing the random variables. Thus, we are considering extending our theoretical and empirical analysis to other estimators such as InfoNCE and its variants for future work. Additionally, in other parts of Appendix B, we discuss other neural network based MI estimators, including compression based and normalizing flow based. We also discuss other pre-processing approaches used in literature in Appendix B, and its potential impact on scale invariance and robustness.
> 2. **Larger datasets:** We have a few bottlenecks currently that prevent us from extending our analysis to larger datasets, particularly KSG. As KSG relies on k-nearest neighbors, the computations quickly become infeasible as either data dimensionality or data size increases beyond the scale of CIFAR-10. Nonetheless, we have now extended our MI vs epoch analysis to the Street View House Numbers (SVHN) dataset. We used approximately 100,000 samples for training, but for the MI Estimation, due to memory issues, we could only use 10,000 samples for estimating MI. The results have been included in Figure 6 of the updated paper. We find that global normalization shows consistent trends for KSG and MINE, and observes a slow compression, whereas other variants are less consistent and do not show a clear trend.

---

### Comment · Action_Editor_L4fF · 2025-05-05
**Discussion**

Dear Reviewers and Authors,

The discussion period has started.

It appears that all reviewers are convinced by the validity of the general direction, though some additional improvements are required to further improve the manuscript. In particular, reviewers tz8y and 4xJv both have concerns about the depth of the experiments, which only use isotropic Gaussians. Can the authors clarify whether using non-isotropic gaussians (or fat tailed distributions) would significantly affect the results?

In addition, I am also particularly interested in the answer to reviewer 4xJv's first question regarding the novelty: are all the propositions in section 3 new? I also encourage the other reviewers to chime in regarding this point and any other points.

Best regards,

AE

---

> ### Author Response · Authors · 2025-05-11
> **Regarding the Queries**
>
> Dear AE,
>
> Yes, we have currently added heavy-tailed distribution using the Student's t-distribution base, and overall the result trends for KSG and MINE estimators seem to be the same as before. In our original experiments, some of our transformation configurations had the cube transformation ($f(x)= x^3$ for each individual dimension), which should yield longer tails, so we were expecting the result trends to not get significantly affected. We have added the new set of results in Table 8 of the revised paper.
>
> Regarding the propositions in section 3, yes, all of them are our results. We have now clarified it in the text.
>
> Best Regards, \
> Authors

---

### Author Response · Authors · 2025-05-11
**Summary of Changes**

Dear Reviewers and AE,

Here is a brief summary of the major additions to our updated manuscript.

1. We have added experiments with the Student's t-distribution base, following the same practices as [1]. (Appendix G, Table 8)
2. We have added an analysis of feature energy versus feature MI to test the assumption for picking global over local normalization (Appendix J, Figure 11)
3. We have added MI versus epoch results for the Street View House Numbers (SVHN) dataset (Figure 6)
4. We have incorporated a summary of discussion of important related works in the main paper (earlier it was in Appendix B alone).

[1]  Czyż, P., Grabowski, F., Vogt, J., Beerenwinkel, N., & Marx, A. (2023). Beyond normal: On the evaluation of mutual information estimators. Advances in Neural Information Processing Systems, 36, 16957-16990.

---

### Decision · Action_Editor_L4fF · 2025-06-14

**Recommendation:** Accept with minor revision

**Additional Comments:**

This paper makes the interesting observation that many mutual information estimators are sensitive to scaling in the data. Theoretically, they show that binning estimators are invariant to scale if we fix the number of bins, but that popular estimators such as KSG or SGD-optimized MINE will return different result if the scale is modified.  The authors then study normalisation schemes and their effect on the accuracy of the Mutual Information estimation task on a wide array of synthetic data involving Gaussians, correlated gaussians, as well as fat-tailed distributions.



Both reviewers tz8y and 4xJv were positive about the paper and its revision since the last submission, which includes many additional experiments. Both reviewers also complained of the lack of experiments with more complex sampling distributions such as fat tailed distributions, a concern which was addressed by the authors by adding many new experiments with student-t distribution. Reviewer 1DxP was more negative, criticising the lack of scope since only two estimators ([MINE] and [KSG]) ares studied and there are no experiments on ImageNet.

Overall, I find the paper very **interesting and deserving of acceptance**. In terms of scope, I think those two estimators are representative enough for this paper to pass the publication threshold, and like reviewers tz8y and 4xJv  I am quite happy with CIFAR+MNIST.



However, I do have some concerns/comments:

[Minor] The authors mention that histogram estimators are not scale sensitive but suffer from the curse of dimensionality. It is definitely worth pointing out that there is a lot of research on structured density estimation which constructs high dimensional histogram-like estimators which do not suffer from the curse of dimensionality. The cost is some assumption in the structure of the ground truth density, with popular choices including a low-rank assumption [LRDE1,LRDE2,LRDE3] or a learnable neural structure [SDE]. The lack of comparison with more subtle structured density estimation approaches is therefore a potential limitation to mention.


[**Major**] Although reviewer 4xJv did mention a general lack of polishing and the presence of many typographical errors, I am surprised none of the reviewers commented more extensively on the mathematical typos.  The **appendix**, and in particular, the **proofs** of the theoretical results **lacks polishing**. I understand that the results are meant to be illustrative rather than constitue true novel theoretical contributions, but I think the proofs still fall short of TMLR standards in terms of mathematical rigour and writing in their current form.

More specifically:

The proofs contain additional “assumptions” which are not always delineated inside the theoretical statements. For instance, the proof of proposition 1 clarifies that the number of bins is assumed to be fixed (rather than the size of the bins), though the estimator is not defined in the statement itself (cf. top of page 34). Much more concerning is the fact that **proposition 5 “assumes” that the weights are bounded throughout the whole optimization procedure**. Although this is indeed written in the proposition statement, this is not an acceptable “assumption” when proving a statement about the gradient trajectory. The fact that the gradient iteration is not used as part of the notation (except for $\delta$, but not for the weights) makes much of the proof difficult to read and makes the statement of the proposition somewhat unclear (the statement holds for a fixed number of gradient steps or epochs, which is not stated in the proposition). In its current form, all the proposition is stating is the following: “Let B be an upper bound on the absolute values of the initial weights. If we train for a fixed number of gradient steps $T$, either  (A) $\lim_{\alpha\rightarrow 0^+} |w_{i,j}(T)|=0$ for all $i,j$ or (B) $\max_{i,j} |w_{i,j}(T)|> B$.” Of course, the initial bound doesn’t need to be tight, so it is reasonable to think we can set $B$ to a higher value to ensure that (B) doesn’t occur, but there is some “homework” to do to rigorously prove that the assumption can be replaced by an assumption at initialization only. The proof contains statements that the intialized weights are "small but nonzero" and appear to be claiming/hinting that this is necessary for the proof to work even though the statement isn't made more precise.




I am recommending **minor revision** because the reviewers’ concerns are all solved, I believe the paper’s overall contribution **merits publication**,  I think it **doesn't make sense to send this paper to another batch of reviewers** and I am happy making the final decision based on the camera ready submission.

*However, I do strongly urge the authors to take the polishing of the proofs of all six propositions seriously to avoid any further delays.*


Additional **minor** typos (non exhaustive):


In equations (14) and (15), I believe the indices are wrong: the first factor of the terms inside the sums should be $\delta^{l}_i (t)$ instead of $\delta^{l}_j (t)$

There are many expressions of the type $h_1,h_2,..,h_{d_n}$ or $h_1,h_2,…h_{d_n}$) instead of $h_1,h_2,\ldots, h_{d_n}$ throughout the text.

Page 35: “unsymmetrical”

Equations (12), (15), (16), (17), (18), and (19) have incorrect punctuation.

Page 17: “representation **of of** the KL-divergence”

Page 16: “many research”, "in $i$th bin$", ...


**References**:

[KSG] W Gao, S Oh, P Viswanath. Demystifying Fixed  k-Nearest Neighbor Information Estimators. TIT 2018
[MINE] Belghazi, M. I., Baratin, A., Rajeshwar, S., Ozair, S., Bengio, Y., Courville, A., & Hjelm, R. D. MINE: Mutual Information Neural Estimation. ICML 2018


[LRDE1] Anandkumar A., Ge R., Hsu D., Kakade S.M., Telgarsky M. Tensor decompositions for learning latent variable models. JMLR 2014.


[LRDE2] R. Vandermeulen and A. Ledent. Beyond Smoothness: Incorporating Low-Rank Analysis into Nonparametric Density Estimation. NeurIPS 2021


[LRDE3] Amiridi M., Kargas N., Sidiropoulos N.D. Low-rank Characteristic Tensor Density Estimation  (Part 1, part 2). TSP 2022.


[SDE] RA Vandermeulen, WM Tai, B Aragam. Dimension-independent rates for structured neural density estimation. ICML 2025.

**Audience:**

Yes

**Audience Explanation:**

Mutual information estimation is of great interest to the community. Further more, reviewers tz8y and 4xJv and myself all enjoyed reading the paper.

**Claims And Evidence:**

Yes

**Claims Explanation:**

Experiments and general idea: yes.

However, the results have some reasonably easily fixable issues that can be addressed during the minor revision stage.